# Persistence and reversal of plasmid-mediated antibiotic resistance

Allison J. Lopatkin[1], Hannah R. Meredith[1], Jaydeep K. Srimani[1], Connor Pfeiffer[2], Rick Durrett[3] & Lingchong You[1,4,5]

In the absence of antibiotic-mediated selection, sensitive bacteria are expected to displace their resistant counterparts if resistance genes are costly. However, many resistance genes persist for long periods in the absence of antibiotics. Horizontal gene transfer (primarily conjugation) could explain this persistence, but it has been suggested that very high conjugation rates would be required. Here, we show that common conjugal plasmids, even when costly, are indeed transferred at sufficiently high rates to be maintained in the absence of antibiotics in *Escherichia coli*. The notion is applicable to nine plasmids from six major incompatibility groups and mixed populations carrying multiple plasmids. These results suggest that reducing antibiotic use alone is likely insufficient for reversing resistance. Therefore, combining conjugation inhibition and promoting plasmid loss would be an effective strategy to limit conjugation-assisted persistence of antibiotic resistance.

[1] Department of Biomedical Engineering, Duke University, Durham, NC 27708, USA. [2] Department of Biology, Duke University, Durham, NC 27708, USA. [3] Department of Mathematics, Duke University, Durham, NC 27708, USA. [4] Center for Genomic and Computational Biology, Duke University, Durham, NC 27708, USA. [5] Department of Molecular Genetics and Microbiology, Duke University School of Medicine, Durham, NC 27710, USA. Correspondence and requests for materials should be addressed to L.Y. (email: you@duke.edu)

Eliminating antibiotic use is an appealing strategy to promote resistance reversal, or the elimination of resistant bacteria by displacing them with their sensitive counterparts[1-3] (Fig. 1a). Indeed, resistance genes often carry a fitness cost, giving the sensitive strains a growth advantage[4-6]. In the absence of selection for antibiotic resistance, competition between the two populations would presumably eliminate the resistant strain over time[5, 7]. However, despite its conceptual simplicity, this approach has been largely unsuccessful[8-10]. Several factors can enable the persistence of resistance in the absence of selection. For instance, co-selection could propagate genetically linked resistance genes[11, 12]. Also, compensatory evolution ameliorating fitness cost can reduce plasmid burden[4, 13, 14].

Horizontal gene transfer (HGT) of plasmids, primarily through conjugation, has also been proposed as a mechanism for plasmid persistence[6, 10, 15]. Theoretical analysis suggests that a sufficiently fast transfer rate can compensate for fitness cost and plasmid loss[16-18], although the extent to which conjugation-mediated maintenance of costly plasmids occurs in nature has been debated[16, 17, 19-22]. For example, it has been suggested that transfer efficiencies required to overcome reasonable estimates of fitness cost and plasmid loss are too high to be biologically realistic[19-22]. Also, the persistence of purely parasitic genetic elements is evolutionarily paradoxical. Overall, conjugation alone is usually not considered to be a dominant mechanism for maintaining plasmids[23-25], although this is not always the case[26, 27].

The fate of a plasmid is largely driven by the relative magnitude of its fitness cost and segregation error rate compared with that of its conjugation efficiency. Indeed, studies investigating conjugal plasmid dynamics in the absence of selection attribute plasmid persistence to fast conjugation rate, low/no fitness cost, or both[22, 28-31]. Similarly, plasmid elimination is attributed to slow conjugation and/or high growth burden[22, 32, 33]. Different outcomes likely depend on underlying parameter differences between experimental systems due to different plasmids, conjugation machinery, and mating procedures, among others.

Accurate quantification of all three processes should provide a general framework to reconcile diverse outcomes and establish the role of conjugation in promoting plasmid persistence. However, confounding measurements of conjugation and growth dynamics have prevented general conclusions[34, 35]. For instance, a high segregation error rate could be obscured by a fast conjugation efficiency[36]. Also, parameters depend on a range of variables including the host strain[37] and growth conditions[32], which complicates data interpretation.

Past studies did not provide precise estimates for all three processes. In some cases, this is because reasonable parameter estimates from available data appear sufficient[22], which is appropriate so long as the estimates are obtained from a relevant experimental framework. In others, experimental complexities prevent accurate quantification within the relevant context (e.g., in vitro growth estimates to evaluate bacterial dynamics in the mouse gut)[29, 32]. Even when all parameters are measured, confounding factors were often not accounted for in the data interpretation. For example, in some cases quantification of the conjugation efficiency was carried out without eliminating the contribution from selection dynamics[32, 33]. As a result, the extent to which conjugation contributes to plasmid maintenance remains inconclusive[6, 23-25, 38, 39].

Lack of basic understanding is prohibitive when evaluating the generality of plasmid fate, such as in natural microbial communities. Typically, microbial consortia consist of multiple interacting populations, connected through a complex network of HGT[6, 40]. High transfer rates in relevant environments (e.g., the gut) have implications in forming communal gene pools where local bacteria can share a wide range (and continuously growing

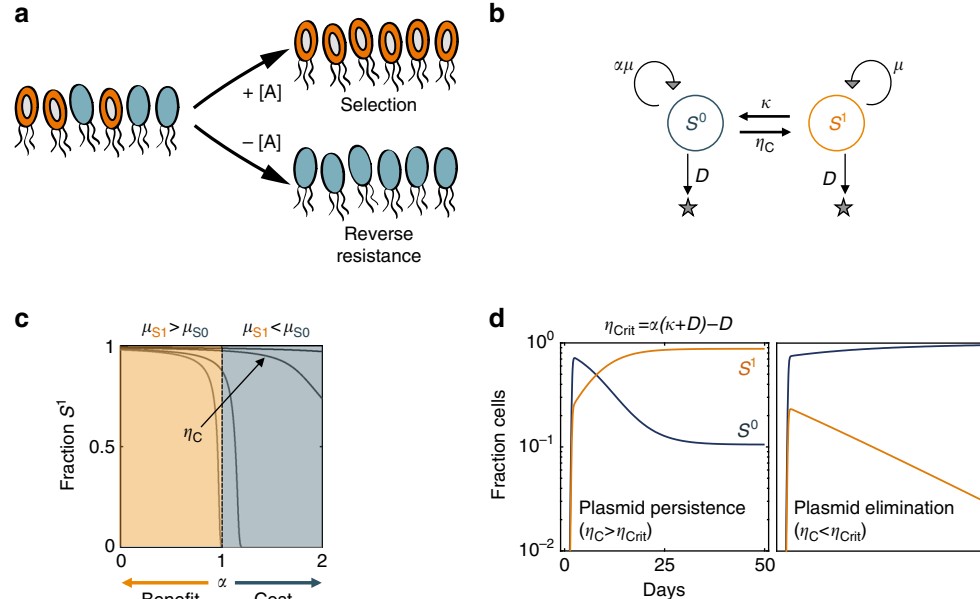

**Fig. 1** Conditions for plasmid persistence and elimination. **a** The concept of resistance reversal. A population initially consists of a mixture of sensitive (blue) and resistant (orange with plasmid) cells. In the presence of antibiotics (± indicates presence or absence of [A] antibiotic concentration), resistant cells are selected for. In the absence of antibiotics, as long as the plasmid imposes a fitness cost, then over a sufficiently long time the resistant cells will be presumably outcompeted, effectively reversing resistance. **b** Modeling plasmid dynamics in a single species ($S$). The plasmid-free population, $S^0$, acquires the plasmid through conjugation at a rate constant $\eta_C$, becoming $S^1$. $S^1$ reverts to $S^0$ through plasmid loss at a rate constant $\kappa$. $S^0$ grows at a rate proportional to $S^1$ ($\mu_1 = \mu$, $\mu_0 = \alpha\mu$). The plasmid is costly when $\alpha > 1$ and beneficial when $\alpha < 1$. Both populations turnover at a constant dilution rate $D$. **c** Simulated fraction of $S^1$ as a function of $\alpha$ and $\eta_C$ after 5000 time units (~200 days). Fast conjugation can compensate for plasmid loss even if the plasmid carries a cost ($\alpha > 1$). A greater $\eta_C$ is required to maintain the plasmid population as $\alpha$ increases. **d** Criterion for plasmid persistence. If $\eta_C > \eta_{Crit} = \alpha(\kappa + D) - D$, the plasmid will dominate (Eq. (1))

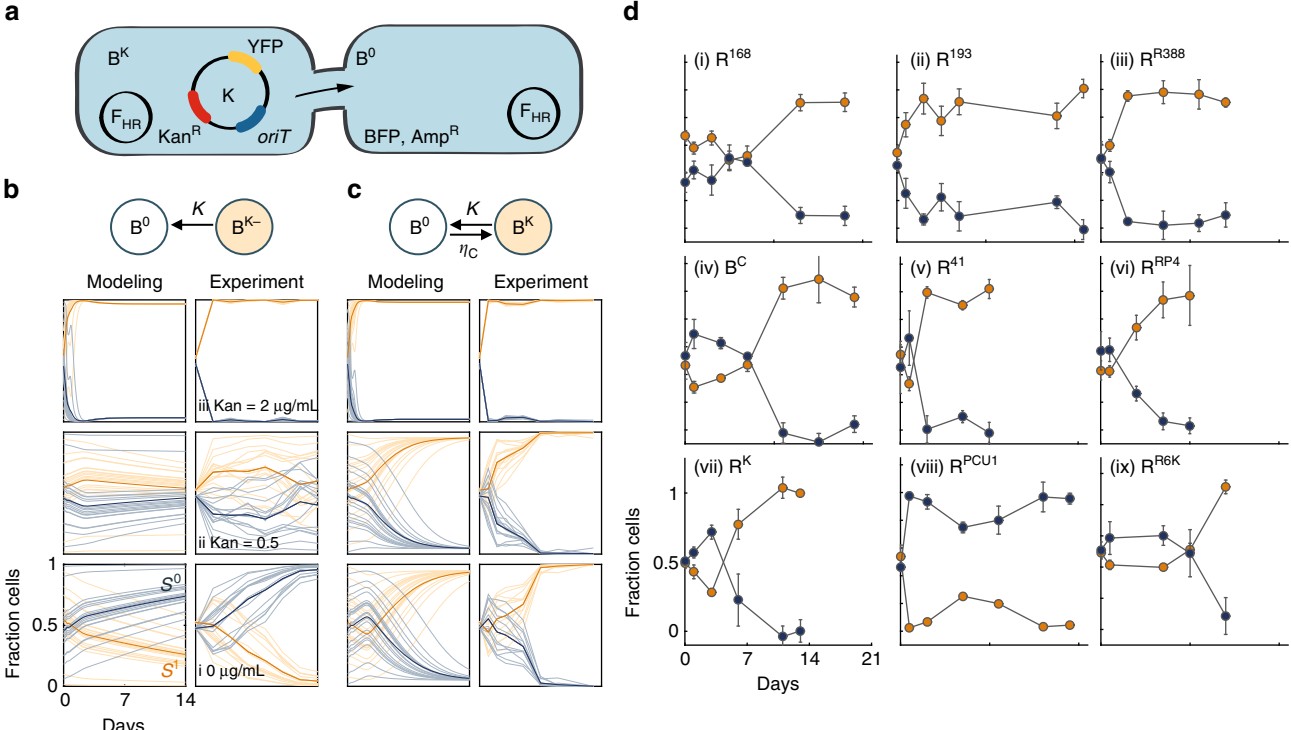

**Fig. 2** Conjugation-assisted persistence of costly plasmids. For all modeling and experimental results, x-axis is days and y-axis is fraction of cells. **a** Engineered conjugation. The background strain, B, expresses BFP and Amp[R] constitutively[45]. B carries the helper plasmid $F_{HR}$ (B[0]), which is non-self-transmissible, but can mobilize plasmids in trans. The mobile plasmid K carries the transfer origin (oriT), a kanamycin-resistant gene (Kan[R]), and yfp under the control of strong constitutive promoter $P_R$[44]. When B carries K, it is denoted B[K]. K without transferability (i.e., without oriT) is denoted K[−], and when carried by B, B[K−]. **b** Long-term dynamics without conjugation. Blue represents plasmid-free and orange plasmid-carrying cells. Shaded lines indicate different initial conditions generated by a strong dilution experimentally (~80 cells/well, 16 wells), or randomly chosen from a uniform distribution (total initial density maintained at $1 \times 10^{-6}$, 20 replicates). Bold lines are the average across all initial conditions of corresponding color. Modeling (left): i–iii is $\alpha =$ 1.02, 0.97, and 0.42, respectively, estimated from experimental measurements (Supplementary Fig. 1C). Experiment (right): i–iii is Kan = 0, 0.5, and 2 µg/mL. Quantification is performed using flow cytometry, where the orange lines are cells expressing both BFP and YFP (B[K−]), and the blue line are cells expressing BFP only (B[0]). **c** Long-term dynamics with conjugation. Experiments were done identically to (B), with B[K] instead of B[K−]. Without antibiotics, the plasmid-carrying population dominated despite the plasmid cost, exhibiting conjugation-assisted persistence. All modeling parameters are identical except for $\eta_C = 0.025$ h$^{-1}$. **d** Nine conjugation plasmids carried by species R (except C with B[0], which behaves similarly, Supplementary Fig. 3D) exhibit conjugation-assisted persistence. R[0] was mixed in equal fraction with R[P] (P for plasmid generality) and diluted 10,000× daily. CFU from four-to-six double-selection plates were divided by the total number of colonies averaged across four-to-six Cm plates for quantification. Experiments are repeated at least twice. Error bars represent the standard deviation of the four-to-six measurements. The plasmids used are (i) #168, (ii) #193, (iii) R388, (iv) C, (v) #41, (vi) RP4, (vii) K, (viii) PCU1, and (ix) R6K (see Supplementary Tables 1 and 3)

pot) of beneficial traits[26]. Disrupting such networks has been recognized as a potential intervention strategy to restore antibiotic susceptibility[41].

Here, we show that common conjugal plasmids covering major incompatibility groups and a range of fitness burdens can be maintained via conjugation, and a simple stability criterion predicts this plasmid persistence. This is true for microbial communities of increasing complexity (e.g., multiple plasmids/strains). Using this framework, we show that reversal or suppression of conjugation-mediated resistance spread is possible by targeting parameters critical for plasmid maintenance.

## Results

**A criterion for plasmid maintenance via conjugation.** We used a simple kinetic model to investigate the extent to which HGT contributes to plasmid maintenance. The model describes one population (S) that either carries the plasmid (S[1]) or is plasmid free (S[0]) (Fig. 1b, Supplementary Methods, Supplementary Eqs. (3)–(4)). In particular, we determined the conditions where the plasmid-carrying population is dominant to provide a conservative estimate for a critical conjugation efficiency. For the

limiting case where the rate of plasmid loss is relatively small (see Supplementary Methods, "Deriving a stability criterion"), we derived a critical conjugation efficiency ($\eta_{Crit}$, Eq. (1)), which approximates an upper bound for dominant plasmid persistence:

$$\eta_{Crit} = \alpha(\kappa + D) - D, \tag{1}$$

where $\alpha$ indicates the relative cost ($\alpha > 1$) or the benefit ($\alpha < 1$) of the plasmid, $\kappa$ is the rate constant of plasmid loss, and $D$ is the dilution rate of the two populations. Thus, we use the term "species" to differentiate between any two populations with a uniquely defined $\eta_{Crit}$, which minimally requires different bacterial clones with genetically distinct backgrounds (e.g., strains or taxonomically diverse species).

According to Eq. (1), a plasmid will be maintained as long as the conjugation efficiency is sufficiently fast compared with the rate of plasmid loss and fitness burden (Fig. 1c, d). Sufficiently fast conjugation efficiency is necessary for the plasmid-carrying population to be dominant (S[1] > S[0], Fig. 1c), even when a plasmid is slightly beneficial ($\alpha$ is slightly <1). Our criterion is similar to that derived by Stewart and Levin[17], but is more stringent in that

it further requires dominance of the plasmid-carrying population. It also avoids experimental challenges associated with decoupling plasmid loss measurements from fitness cost[42]. Experimentally, observed plasmid loss ($\kappa_{obs}$) can be determined by measuring the time constant of decay for a non-transferrable plasmid, which represents a combined effect of true $\kappa$ and $\alpha$ (Supplementary Fig. 1A)[43]. Indeed, analysis shows that $\kappa_{obs} \approx \kappa$ for plasmids with minimal fitness effects ($\alpha \approx 1$). Since these two parameters are challenging to decouple, our criterion lumps the effects of $\alpha$ and $\kappa$ together. To determine $\kappa_{obs}$, we chose to use a low-cost plasmid ($\alpha = 1.02$) to minimize the confounding effects of cost. Based on our experimentally determined parameters, analysis shows the standard error associated with fitting the plasmid loss rate ($\approx 0.0022$) is greater than the difference between $\kappa_{obs}$ and $\kappa$ (see Supplementary Methods, "Plasmid loss calculations", for complete derivation).

**Conjugation-assisted persistence in a synthetic system.** To test whether the conjugation efficiency for common conjugal plasmids is sufficiently fast to compensate for cost, we first adopted a synthetic conjugation system derived from the F plasmid. In this system, the conjugation machinery is encoded on a helper plasmid $F_{HR}$, which is not self-transmissible[44]. A second plasmid can be mobilized through conjugation when it carries the F origin of transfer sequence *oriT*. Here, we use a mobilizable plasmid denoted K, which expresses YFP under the control of the strong constitutive promoter $P_R$, a kanamycin (Kan) resistance gene ($kan^R$)[44], and *oriT*. To quantify the effects of conjugation, we implemented a plasmid identical to K except that it does not carry *oriT* (K⁻), and therefore cannot be transferred by conjugation. The synthetic system was introduced into an engineered derivative of *Escherichia coli* MG1655 expressing a constitutive blue fluorescent protein (BFP) chromosomally and carbenicillin (Carb) resistance (Amp$^R$)[45], denoted B (Fig. 2a). The plasmid-carrying populations (B$^K$, B$^{K-}$) can be distinguished from the plasmid-free population (B$^0$) by selective plating (using Carb +Kan) or flow cytometry (using YFP) (Supplementary Fig. 2A). This notation will be used to describe all species and plasmid combinations throughout the text (see Supplementary Methods, "Nomenclature").

This system provides a clean experimental configuration to elucidate the contribution of conjugation to plasmid maintenance. K enables more precise parameter estimates compared to natural self-transmissible plasmids. Native plasmids often encode additional functions that complicate measurements, such as addiction modules that can result in post-segregational killing of daughter cells[46]. Importantly, without a non-transmissible control plasmid, it is difficult to decouple the effects of HGT from other processes. Instead, plasmid loss and fitness burden can be precisely quantified using K⁻, which eliminates the confounding influence of conjugation. Since *oriT* did not significantly affect the burden of K compared to K⁻ (Supplementary Fig. 1B, $P > 0.5$, two-sided *t*-test), differences that arise in the overall dynamics can be attributed to conjugation.

From our measurements of K, we expect conjugation to be fast enough to enable maintenance in the absence of antibiotic selection. In particular, our measurements of $\kappa = 0.001$ h$^{-1}$ (Supplementary Fig. 1A), $\alpha = 1.02$ (Supplementary Fig. 1C), and assuming $D \approx 0.05$ h$^{-1}$, we estimate the critical efficiency $\eta_{Crit} = 0.002$ h$^{-1}$ to be well below the estimate of conjugation efficiency from exponential phase growth $\eta_C \approx 0.01$ h$^{-1}$[34] (see Methods, "Estimating $\eta_{Crit}$"). Indeed, cell physiology can drastically change the conjugation efficiency[34], as this value is almost four orders of magnitude greater than the efficiency measured from cells harvested from stationary phase (Supplementary Table 3).

To test conjugation-mediated plasmid maintenance, we mixed B$^K$ and B$^0$ in equal fractions and cultured them together. A strong dilution (10,000×) was performed every 24 h to maintain growth. Different concentrations of Kan (0, 0.5, and 2 µg/mL) were used to vary $\alpha$ (1.02, 0.97, 0.42, respectively) (Supplementary Fig. 1C). Every few days, we quantified the fractions of plasmid-bearing cells (expressing BFP and YFP) and plasmid-free cells (expressing BFP only) using flow cytometry (see Supplementary Fig. 2A and Methods section "Flow cytometry calibration" for calibration details).

In the absence of conjugation, when the plasmid carries a cost (e.g., Kan = 0 and $\alpha > 1$) the plasmid-bearing population was eliminated after 2 weeks (Fig. 2b, left modeling and right experiment). Thus, for a non-transferrable costly plasmid, eliminating antibiotics results in resistance reversal. If the plasmid was sufficiently beneficial, the plasmid-bearing population could coexist with the plasmid-free population (Fig. 2b, left modeling and right experiment). The fraction of plasmid-bearing cells depended on the relative magnitude of growth advantage compared with plasmid loss. In contrast, if the plasmid is transferrable through conjugation, even when the plasmid carries a cost, plasmid-bearing cells dominate the population in a short period of time (Fig. 2c, left modeling and right experiment). Intuitively, decreasing cost or increasing benefit (i.e., decreasing $\alpha$) facilitates conjugation-assisted persistence, and therefore plasmid stability occurs on a faster timescale (Fig. 2c). Once the plasmid benefit is sufficiently high (Fig. 2c), the plasmid persists regardless of whether or not it can conjugate, indicating that conjugation is no longer required to maintain resistance.

Further, analysis suggests compensatory mutations, even at a high mutation rate, did not contribute significantly to the overall dynamics (Supplementary Fig. 2B). We note that our model assumes a constant dilution rate constant ($D$), which represents an approximation of the discrete, periodic dilutions in our experiments. Simulations using a model implementing discrete dilutions generated qualitatively the same results (Supplementary Fig. 1D). Finally, we introduced noise in the conjugation rate for each set of initial conditions such that $\eta_C$ can vary a small amount from the basal value, within 10% of the mean, consistent with clonal variability[34]. This variability does not change the qualitative results (Supplementary Fig. 2C).

**Conjugation-assisted persistence for diverse plasmids.** We previously demonstrated that the conjugation efficiency of the synthetic system is comparable to that of natural F plasmids and several other natural conjugation plasmids[34]. Therefore, we expect these plasmids to also exhibit conjugation-assisted persistence. To this end, we quantified the dynamics of eight additional conjugative plasmids, covering six incompatibility groups (incF, incN, incI, incX, incW, and incP) which encompass >70% of the most common large plasmids isolated from *Enterobacteriacea* (335 plasmids that are >20 kB from GenBank)[47], cover a wide range of conjugation efficiencies and fitness effects (Supplementary Fig. 3A, B), and include three clinically isolated conjugative plasmids encoding extended-spectrum β-lactamases (ESBLs). ESBL-producing pathogens are notorious for plasmid-mediated conjugation[48–50] and are of paramount global health concern[51, 52]. We transferred each individual plasmid into a common background strain (*E. coli* MG1655 with chromosomally integrated dTomato, and chloramphenicol (Cm) resistance (Cm$^R$), denoted R), and quantified the relevant parameters to estimate $\eta_{Crit}$; the plasmid C was quantified with background strain B since both plasmid C and strain R express Cm$^R$, and B behaves qualitatively similarly to R (Supplementary Fig. 3D).

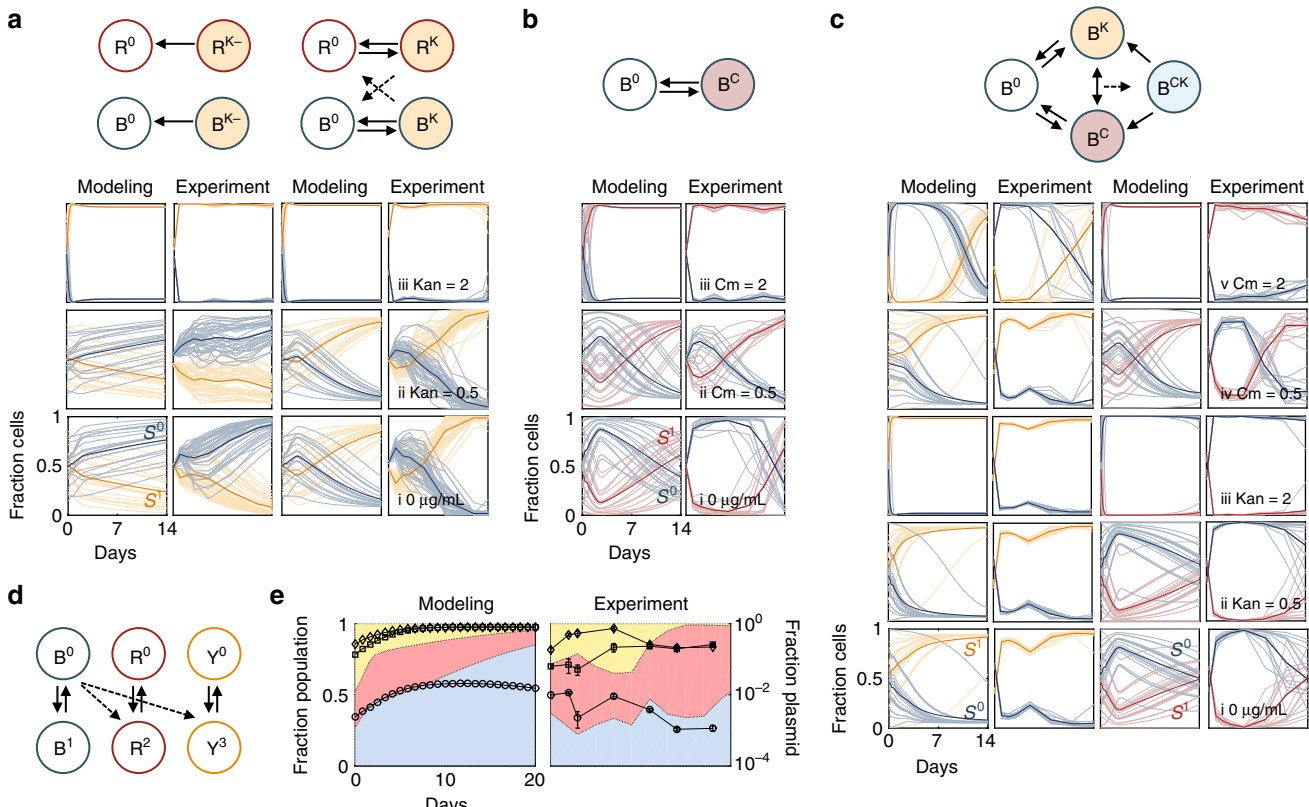

**Fig. 3** Conjugation-assisted persistence with multiple species and/or plasmids. **a–c** x-axis is days and y-axis is fraction of cells. Bold and shaded lines represent average across, or individual, initial conditions, respectively. Color indicates blue for plasmid free ($S^0$), and orange or red for plasmid-carrying cells ($S^1$) K or C, respectively. **a** Two-species, one-plasmid community. Left two panels: no conjugation; right two panels: with conjugation. $S^0 = B^0 + R^0$ and $S^1 = B^K + R^K$. Modeling: From bottom (i) to top (iii) $\alpha_1 = \alpha = 1.02$, 0.97, and 0.42, respectively, and $\alpha_2 = 1.03$, 1.02, and 0.9 (see Supplementary Eqs. (7)–(10), Supplementary Fig. 4A). Experiment from bottom (i) to top (iii): Kan = 0, 0.5, and 2 μg/mL, respectively. **b** Higher cost plasmid dynamics. Modeling (left column): From bottom (i) to top (iii) $\alpha = 1.13$, 1.03, and 0.3, respectively (see Supplementary Eqs. (3)–(4), Supplementary Fig. 4B). Experiment (right column): From bottom (i) to top (iii): Cm = 0, 0.5, and 2 μg/mL. **c** One species, two-plasmid community. Each row represents a different combination of $\alpha$, modulated with no antibiotic (i), Kan (ii–iii), or Cm (iv–v). The species can carry two ($S^{11}$), one ($S^{10}$, $S^{01}$), or no plasmids ($S^{00}$). Modeling (first and third columns): From bottom (i) to top (v) $\alpha_3 = 1.3$, 1.2, 0.42, 1.01, 0.35 (see Supplementary Eqs. (11)–(14), Supplementary Fig. 4B). Experiment: (second and fourth column such that $S^1 = B^K + B^{CK}$ or $S^1 = B^C + B^{CK}$ for K or C, respectively). $B^C$ is mixed equally with $B^K$. **d, e** Three-species, three-plasmid community. Species (R, Y, and B) are uniquely fluorescent (expressing dTomato, YFP, or BFP, respectively) and plasmids (R6K, RP4, and R388, diamond, square, and circle markers, respectively) have distinct resistance markers (Strp$^R$, Kan$^R$, and Tm$^R$, respectively). Shading color corresponds to the respective population fraction (left y-axis), and markers indicate fraction of each plasmid (right y-axis). The initial experimental composition consists of R$^0$, R$^{R6K}$, Y$^0$, Y$^{R388}$, B$^0$, and B$^{RP4}$. Modeling (left): Randomized initial conditions such that the total plasmid-free populations is maintained at $1 \times 10^{-4}$, and plasmid population arbitrarily chosen between $1 \times 10^{-5}$ and $1 \times 10^6$, consistent with data (Supplementary Table 2 for parameter estimates). Experiment (right): Error bars indicate averaging across four-to-six plate replicates, and repeated five times

Our estimates suggest a high likelihood for persistence ($\Delta n = \eta_C - \eta_{\text{Crit}} > 0$) for each of the nine plasmids (Supplementary Fig. 3C, including R$^K$ for control), either because they are sufficiently beneficial and/or transferred fast enough. To test this, we implemented the same competition experiments as previously described, and quantified the fraction of plasmid-bearing cells using colony-forming units (CFU) on double-antibiotic plates (see Methods). Daily dilutions were performed for 14–20 days. Indeed, each plasmid persisted throughout the duration of the experiment (Fig. 2d). The maintenance or dominance of several plasmids (#168, #193, RP4, R6K, and R388) was likely due to them being neutral or slight beneficial (Supplementary Fig. 3C), in addition to their fast transfer. In contrast, PCU1 was maintained despite its very high cost (estimated $\alpha \approx 3$, Fig. 2d; see Supplementary Fig. 3E for logscale).

**Conjugation-assisted persistence in greater complexity consortia.** Natural environments are typically far more complex, consisting of diverse species interconnected through an intricate

web of gene exchange[6, 53]. Such networks can serve as reservoirs for antibiotic resistance in so-called HGT "hot spots", enabling the dissemination of resistance to various pathogens or commensal microbes[54–56]. Therefore, we wondered whether conjugation-assisted persistence could occur in a multi-species community. This question was never conclusively explored previously.

Modeling suggests that, as long as the stability criterion is met, a single plasmid can be maintained via conjugation regardless of the number of species present (Fig. 3a, Supplementary Methods, Supplementary Eqs. (7)–(10)). To test this, we introduced a second *E. coli* strain R with or without *oriT* (R$^K$ or R$^{K-}$) (Supplementary Fig. 4A). The total plasmid content is quantified as the sum of all plasmid-bearing species (R$^K$+B$^K$). Consistent with our predictions, results demonstrate that conjugation enables plasmid persistence compared to the non-conjugating control (Fig. 3a).

Moreover, modeling predicts conjugation-assisted persistence to occur for a single species carrying multiple conjugation

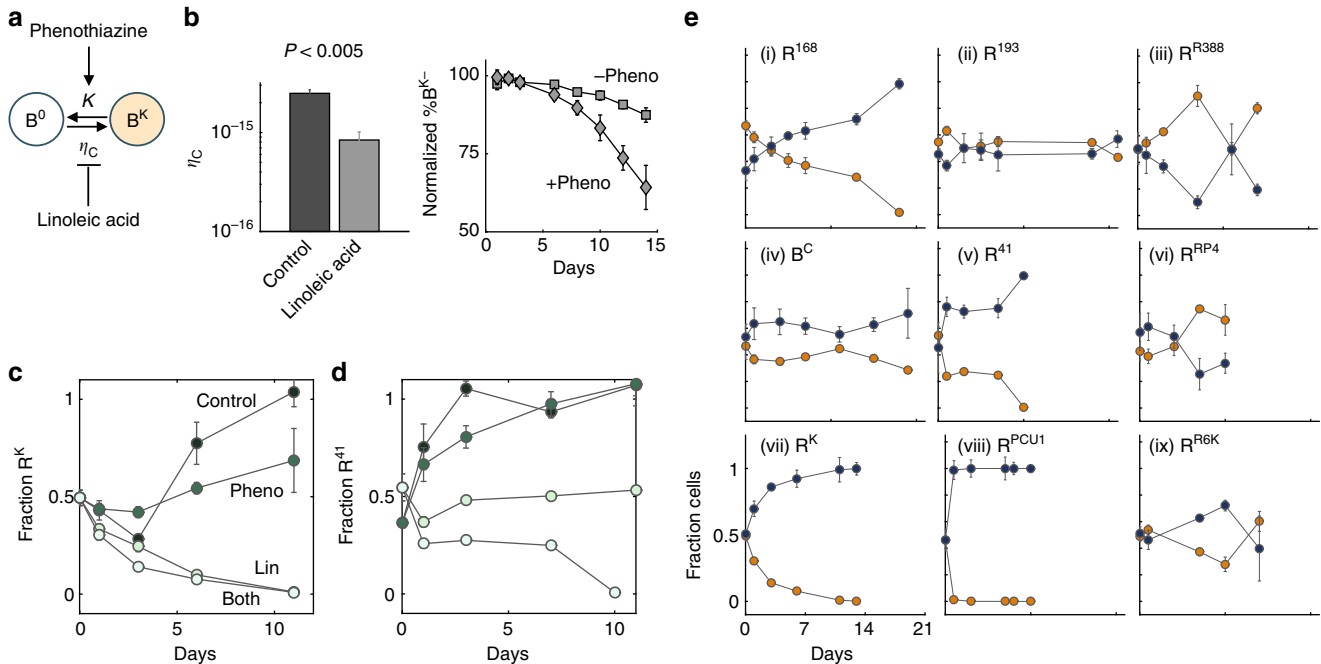

**Fig. 4** Reversing resistance due to conjugation-assisted persistence. **a** Combining inhibition of conjugation and promotion of plasmid loss to reverse resistance. This strategy is expected to increase $\eta_{\text{Crit}}$ and decrease $\eta_C$, potentially destabilizing the plasmid (Eq. (1)). **b** Evaluating conjugation inhibitor linoleic acid (Lin) and plasmid loss rate promoter phenothiazine (Pheno). Left: $B^K$ and $R^0$ were grown overnight with or without 3.25 mM Lin to quantify conjugation efficiency (see Methods). Right: $B^{K-}$ was propagated daily in the presence of 50 µg/mL Kan, nothing, or 120 µM of Pheno. Kan was used as a control. $Y$-axis is the fraction of $B^{K-}$ without antibiotic normalized by that treated with Kan quantified via flow cytometry. Pheno significantly increased the rate of plasmid loss by ~four-fold (see Supplementary Fig. 6B, right panel). **c, d** Inhibition of $R^K$ and $R^{41}$. $R^0$ and $R^K$ or $R^{41}$ were mixed in equal fractions and diluted 10,000× daily for 11 days. $Y$-axis is fraction of plasmid-carrying cells and $x$-axis is days. Green shading indicates the treatments from dark to light: control, Pheno, Lin, and combined. Both plasmids were successfully reversed; when Lin was sufficient alone, Pheno had minimal effect (K). If Lin alone was insufficient, Lin with Pheno synergistically destabilized the plasmid. **e** Combination treatment with Lin and Pheno suppressed or reversed resistance. The same strains and protocol were used as in Fig. 2d, except media was supplemented with 3.25 mM Lin and 120 µM Pheno fresh daily (see Methods). The plasmids used are (i) #168, (ii) #193, (iii) R388, (iv) C, (v) #41, (vi) RP4, (vii) K, (viii) PCU1, and (ix) R6K (see Supplementary Tables 1 and 3). All CFU measurements were done in replicates of four-to-six plates, and repeated at least twice for reproducibility. All flow measurements were propagated with at least eight well replicates and repeated at least twice for reproducibility. Error bars represent the standard deviation of the plate or well replicates

plasmids. This is contingent on the plasmids' ability to exist independently of each other (e.g., distinct incompatibility groups to ensure compatible replication machinery and the absence of surface exclusion that prevents entry of one of the plasmids), and the fact that other relevant plasmid parameters (i.e., $\eta_C$, $\alpha$, or $\kappa$) are not drastically altered by the presence of another plasmid (Supplementary Methods, Supplementary Eqs. (11)–(14)). To test this idea, we implemented a bi-directionally conjugating population by mixing B carrying either K or another mobilizable plasmid C. C expresses mCherry, $\text{Cm}^R$, and is compatible with K (p15A and pSC101 replication origins, respectively). Independently, since C has a greater cost compared to K (Supplementary Fig. 3B), conjugation required a longer time-scale to overcome competition and stably persist (Fig. 3b). Together, the dynamics of each plasmid individually were identical to that of the single-plasmid population dynamics, regardless of how we modulated $\alpha$ (Fig. 3c, no antibiotic, Kan, and Cm; see Supplementary Fig. 4A, B for $\alpha$ estimates).

These results suggest that, despite the apparent complexity, plasmid fate in a community consisting of multiple ($n$) species and ($p$) plasmids (leading to $n2^p$ populations) can be inferred from the individual plasmid dynamics, if these plasmids do not interfere with each other. The fate of each plasmid is governed by the criterion for conjugation-assisted persistence (e.g., $\eta_C > \eta_{\text{Crit}}$). If $\eta_C > \eta_{\text{Crit}}$ for at least one such pair, the plasmid will persist if the particular host(s) can coexist within the population long term

(which is largely driven by fitness). Importantly, the coexisting species must acquire the plasmid, either in the initial population structure, or via conjugation. This results in an initial barrier for the plasmid to establish itself due to competition, resulting in a dependence on the initial composition in determining plasmid fate (see Supplementary Methods, "Three-species three-plasmids model").

To test this, we constructed a community consisting of three species (*E. coli* strains denoted $B^0$, $R^0$, and $Y^0$) transferring three mutually compatible plasmids (Supplementary Fig. 5A, Supplementary Methods, Supplementary Eq. (16), Supplementary Table 4). Each plasmid was initiated in a single species, respectively (RP4, R6K, and R388, denoted 1, 2, and 3 in Fig. 3d, Supplementary Fig. 5A). These three plasmids were chosen in particular since they belong to distinct incompatibility groups (X, P, and W), and are distinguishable using antibiotic selection (Streptomycin (Strep), Kan, and Trimethoprim (Tm), respectively, Supplementary Table 3). Since all species express chromosomal $\text{Cm}^R$, we used selective plating to determine the plasmid fraction and flow cytometry to determine the species composition. In this scenario, although plasmids individually appear beneficial to their own host ($\alpha < 1$), they exhibit a cost compared to another species/plasmid pair (e.g., compare $B^{RP4}$ to $R^{R388}$, Supplementary Fig. 5B). Based on our previous estimates, we predict persistence for all three plasmids in this community. Indeed, results demonstrate that each individual plasmid

exhibited persistence throughout the duration of the experiment, for up to 2 weeks (Fig. 3e). In the absence of conjugation ($R^0$, $B^0$, and $Y^0$ only), competition between the three species favors the fittest population ($Y^0$), suppressing the growth of both $R^0$ and $B^0$ (Supplementary Fig. 5B).

**Reversing conjugation-assisted persistence of resistance.** Our results demonstrate that diverse conjugal plasmids are indeed transferred fast enough to enable plasmid persistence. According to the existence criterion (Eq. (1)), however, resistance reversal can be achieved by inhibiting conjugation, promoting the rate of plasmid loss, or both (Fig. 4a). The efficacy of this strategy depends on how much $\eta_C$ exceeds $\eta_{Crit}$. If $\eta_C$ is only slightly greater than $\eta_{Crit}$, inhibiting conjugation alone might be sufficient to reverse resistance. If inhibition alone is incomplete, however, promoting $\kappa$ may act in synergy to destabilize the plasmid.

We first tested this inhibition strategy on the engineered conjugation system by using linoleic acid[57] (Lin) to inhibit conjugation (Fig. 4b, left panel) and phenothiazine (Pheno) to enhance the plasmid segregation error[58, 59] (Fig. 4b, right panel). Both compounds had been identified in literature for these specific properties. Importantly, at the concentrations we used, neither compound affected the bacterial growth rate (Supplementary Fig. 6A). Indeed, Lin alone was sufficient to destabilize a plasmid with low conjugation efficiency (Fig. 4c, plasmid K). For a plasmid with greater $\eta_C$ (e.g., for plasmid #41) or conferred a benefit, Lin alone was insufficient, and the synergistic combination of Lin with Pheno was critical to reverse resistance (Fig. 4d). We note that Pheno alone did not affect the conjugation efficiency (Supplementary Fig. 6B).

We found that Lin reduced the conjugation efficiency for most of the native plasmids by three-fold (Supplementary Fig. 6C) and even by 50-fold in one (see Supplementary Table 3 for all fold changes). Adjusting for this decrease in predicted $\eta_{Crit}$, maintaining the same cost (Supplementary Fig. 6D), and assuming a four-fold increase in the Pheno-enhanced plasmid loss rate (Supplementary Fig. 6B, right), our criterion predicts that conjugation-assisted persistence would be significantly reduced for most plasmids (Supplementary Fig. 6E, Supplementary Table 3). Indeed, a combination of Lin and Pheno led to >99% elimination of plasmids where $\Delta n < 0$ (Fig. 4e, plasmids K, #41, #168, PCU1, Supplementary Fig. 6F for log-scale PCU1 as comparison to Supplementary Fig. 3E). If $\Delta n$ is close to but slightly greater than 0, the plasmid still persisted but with a reduced infectivity (Fig. 4e plasmids C, #193). If $\Delta n$ is sufficiently large (>>0), the plasmid was maintained (Fig. 4e plasmids RP4, R6K, R388). However, they were less dominant in comparison with the absence of Lin and Pheno (Fig. 2d), indicating the role of conjugation in their maintenance. That these three plasmids were more difficult to reverse is not surprising, since they carry a small burden or even a benefit, to R (Supplementary Fig. 3B).

## Discussion

It is estimated that >50% of all plasmids can be transferred by conjugation[25]. The extent to which conjugation contributes to the difficulty to reverse antibiotic resistance has been debated for decades[16, 38, 60]. Some have suggested that conjugation is not a major mechanism responsible for the persistence of plasmids[20, 23, 39, 61]. We believe that the root of this apparent confusion is the lack of precise quantification of the plasmid dynamics as affected by conjugation, under environments with varying degrees of selection. Moreover, how conjugation contributes to plasmid persistence in a generic multi-species community has not been previously investigated.

Our results demonstrate that the transfer of various conjugal plasmids is sufficiently fast to exhibit persistence. This is consistent with recent work demonstrating the persistence of heavy metal resistance in the absence of positive selection when transferrable via conjugation, compared to inheritable solely through clonal expansion (the latter of which required positive selection to maintain)[62]. The plasmids tested here cover six of the major incompatibility groups. Many plasmids encoding ESBL genes spread to diverse species through conjugation[48, 63]. Our findings may shed light on the apparent paradox regarding the high prevalence of ESBL resistance, despite studies having determined ESBL plasmids are often costly to maintain[33, 64, 65]. Indeed, even in the absence of β-lactam treatment, our findings suggest that ESBL resistance is likely to persist for long periods of time. We further demonstrated that conjugation-assisted persistence is generally applicable to communities containing multiple plasmids and multiple populations each defined by a unique critical conjugation efficiency. This is particularly relevant, since HGT is pervasive, and extensively occurs in hot spots associated with highly dense and complex population structures, such as in the gut microbiome[54]. That independent plasmid dynamics are additive greatly simplifies our understanding of plasmid dynamics in populations with greater complexity, as well as future work investigating intervention strategies.

Several other factors may contribute to the long-term stability of such plasmids in the environment. Indeed, almost half of all plasmids are unable to transfer via conjugation. Co-evolution between the host and plasmid can compensate for fitness cost[13, 14, 24, 66, 67], and recent work has shown that other factors such as positive selection coupled with compensatory adaptation can help explain long-term plasmid persistence[24, 68]. Interestingly, compensatory mutations may also modulate other key processes involved with mobile genetic element (MGE) upkeep by directly or indirectly modulating processes involved in conjugation, such as decreased expression of MGE replication[67], down regulation of global gene expression[69], or increased plasmid copy number[68]. Indeed, recent theoretical work emphasized the potentially paradoxical role of these interacting processes, where reducing fitness cost could mitigate the evolution for higher conjugation rates[70]. Future work investigating the extent to which compensatory mutations individually modulate $\alpha$, $\kappa$, $\eta_C$, or some combination thereof, would provide critical insight into predicting evolutionary trajectories that enhance plasmid stability.

Population dynamics are another potential contributing factor; one study showed that the presence of alternative hosts can promote the survival of a plasmid unable to persist in mono-culture[71]. The physical environment may play an important role in plasmid persistence as well, since HGT dynamics change depending on the spatial structure of the community (reviewed in ref. [6]).

Our findings demonstrate the necessity to inhibit the conjugation process for effective resistance reversal. In particular, in the absence of active intervention strategies, it is likely that judicious antibiotic use may only suspend the process of continued selection and enrichment, but would be unable to reverse resistance due to conjugation[72, 73]. To this end, the synergy between plasmid-curing compounds and conjugation inhibitors represents a novel approach for antibiotic adjuvants aimed at targeting the ecological and evolutionary aspects of bacterial pathogenesis[3, 41].

## Methods

**Strains, growth conditions, and plasmid construction.** Different *E. coli* strains were used throughout the study (Supplementary Table 1). Derivatives of *E. coli* strain MG1655 with chromosomal fluorescence and antibiotic resistance were generously provided by the Andersson lab[74]. Recipients $B^0$ and $R^0$ each carry

helper F plasmid $F_{HR}$, express BFP and ampicillin (Amp, or Carb for carbenicillin) resistance ($Amp^R$), or dTomato and $Cm^R$, respectively. Note that for the multi-plasmid multi-species experiment, $B^0$ expresses $Cm^R$ instead of $Amp^R$ (see Supplementary Table 1 for a complete list of strain details). Donor cells ($B^0$ or $R^0$ background) contain mobilization plasmid K from Dimitriu et al. (denoted Y in referenced publication)[44], which carries *yfp* gene under the control of strong constitutive $P_R$ promoter, *oriT* for transfer, and expresses kanamycin (Kan) resistance ($Kan^R$) (denoted $B^K$). For conjugation controls, non-transferrable plasmid $K^-$ is used, which is identical to K but without *oriT*. Upon conjugation, transconjugants become indistinguishable from the donors. For experiments with a more costly plasmid (Fig. 3b), plasmid C is used. C expresses $Cm^R$, mCherry, and has p15A replication origin. For a complete list of strains and plasmids used in this study, see Supplementary Table 1. For all experiments, single clones were grown separately overnight at 37 °C for 16 h with shaking (250 rpm) in Luria-Bertani (LB) broth containing appropriate antibiotics (100 µg/mL Cm, 100 µg/mL Carb, or 50 µg/mL Kan). All experiments were performed using M9 medium (M9CA medium broth powder from Amresco, lot #2055C146, containing 2 mg/mL casamino acid, supplemented with 0.1 mg/mL thiamine, 2 mM $MgSO_4$, 0.1 mM $CaCl_2$, and 0.4% w/v glucose).

**Long-term plasmid dynamics for engineered conjugation.** 16 h overnight cultures (3 mL LB media with appropriate selecting agents, density ~$1 \times 10^9$ CFU/mL) were resuspended in M9 medium and diluted to an initial starting density of ~80 cells/well. Strong initial dilutions were used to generate replicates with a range of initial conditions per well. Depending on the experiment, replicates ranged from 12 to 48 wells. For the one-species one-plasmid conjugation experiment (Fig. 2c), $B^0$ and $B^K$ cells were mixed in an equal ratio and strongly diluted to an initial density of ~80 cells/well combined. The strongly diluted cell mixture was distributed among 24-well replicates in a 96-well plate to a final volume of 200 µL/well, supplemented with appropriate antibiotic (Kan = 0, 0.5, or 2 µg/mL). 96-well plates were covered with an AeraSeal™ film sealant (Sigma-Aldrich, SKU A9224) followed by a Breath-Easy sealing membrane (Sigma-Aldrich, SKU Z380059). Plates were shaken at 250 rpm for 23 h and 37 °C. This is denoted Day 0.

To passage the plates, 198 µL/well of freshly mixed media was distributed into a new 96-well plate, and an intermediate passaging plate containing 198 µL/well of autoclaved diH$_2$O was prepared. 2 µL from the previous day's plate was transferred to the intermediate passaging plate, mixed by pipet, and then 2 µL from the intermediate passaging plate was transferred into the new media-containing plate to achieve daily dilutions of 10,000×. The new plate was sealed using both membranes, and placed back into the incubator to shake. This process takes approximately 1 h. An additional 2 µL of cells was added to the intermediate plate to be used for flow cytometry, and the experiments were carried out typically for 14–18 days.

The same protocol is used for the one-plasmid two-species experiment (Fig. 3a, 48-well replicates), testing the additional, more costly, plasmid C (Fig. 3b, 12-well replicates), and the two-plasmid one-species experiment (Fig. 3c, 12-well replicates). All experiments are initiated with equal volumes of all populations, except the latter, which initiates with only the two populations carrying each plasmid ($B^K$ and $B^C$). The initial starting density was maintained at ~80 cells/well for each of these three experiments as well. Non-conjugating experiments had an identical setup with the substitution of $K^-$ plasmid for K.

**Flow cytometry calibration.** To calibrate flow cytometry for accurate quantification of plasmid-free and plasmid-carrying populations we mixed various ratios of the cell populations $B^0$, $R^0$, $B^{K-}$, and $R^{K-}$ in different combinations with one another (Supplementary Fig. 2A). Volume ratios were used as a proxy for cell ratios since differences in overnight CFU for all populations was statistically indistinguishable ($P > 0.5$, two-sided *t*-test). Cell mixtures were calibrated at high (400×) and low (1000×) dilution-fold from overnight culture, resulting in a flow speed of between 3000–6000 cells/s. Fluorescent gating cutoffs were determined in such a way that were both obvious by eye, and resulted in <5% differences between the population mixtures (as determined by volume) when quantified. 5% was chosen as this is the detection limit for the machine. All data quantification was performed using MATLAB. The same threshold values were used to gate every experiment. Flow measurements were always performed on live cells, in diH$_2$O, on the actual day of the experimental time point, within 1 h of removing the plate from the incubator.

**Quantifying modeling parameters $\alpha$, $\eta$, $\kappa$.** $\alpha$ was determined using plate-reader measurements of B and R growth in the presence of 0, 0.5, or 2 µg/mL Kan using the Perkin-Elmer Victor ×3. Overnight cultures of cells (prepared as described previously) were resuspended into M9 media and diluted 10,000× prior to starting the experiment. Four technical replicates per concentration were used for quantification. Growth rates were quantified by log-transforming the growth curves, using K-means clustering to non-arbitrarily locate the region of longest exponential growth, smoothing, and fitting the linear portion (all processing done using MATLAB). $\alpha$ is determined by normalizing the growth rate obtained from the plasmid-free population ($R^0$ or $B^0$) by the plasmid-carrying population ($R^K$ or $B^K$). These experiments were performed at three antibiotic concentrations used in the

main experiments to accurately model the individual dynamics (Supplementary Figs. 1C, 4A, B). The growth temperature used to quantify $\alpha$ corresponds to that which was used for long-term experiments (e.g., 37 °C for R/$R^K$, and 30 °C for R/$R^{RP4}$, see Supplementary Table 1 for each specific condition).

Conjugation efficiency was estimated using the protocol established by Lopatkin et al[34]. 16 h overnight cultures of $B^K$ and $R^0$ (3 mL LB media with appropriate selecting agents, density ~$1 \times 10^9$ CFU/mL) were resuspended in M9 medium (0.4% w/v glucose) and mixed in a 1:1 ratio using 400 µL. Mixtures were incubated at room temperature (25 °C) for 1 h in the absence of shaking. $R^0$ and $B^K$ are used pairwise in these experiments to take advantage of the different resistant markers: Kan (50 µg/mL) is used to quantify donors, Cm (100 µg/mL) for recipients, and the Kan+Cm combination can distinguish transconjugants. Parental densities were obtained by serially diluting the mixture to net $5 \times 10^7$-fold, and spreading four-to-six replicate measurements onto respective single-antibiotic-containing plates. To quantify transconjugants ($R^K$, as a result of transfer from B to R), cells were plated at a dilution of 40-fold onto plates containing both antibiotics. Plates were incubated overnight at 37 °C and CFUs were counted the following day.

The conjugation efficiency is estimated as $\eta = \frac{R^K}{B^K R^0 \Delta t}$, with units of cell mL$^{-1}$ h$^{-1}$, and is dependent on the cell density. For all modeling and plasmid persistence predictions, we use the cell density-independent rate constant $\eta_C$, by normalizing $\eta$ with respect to the carrying capacity. That is, we let $\eta_C = \eta N_m$, where $N_m = 10^9$ cell/mL (the carrying capacity). $\eta_C$ has units of h$^{-1}$.

To estimate the plasmid loss rate, experiments were initiated with 100% of $B^{K-}$, the non-transferrable variant of K. Cells were diluted 10,000× daily and monitored for approximately 1 week, at which point sufficient plasmid loss was observed (Supplementary Fig. 1A) such that exponential functions could be fit to the loss curves. The percentage $B^{K-}$ ($B^{K-}$ with no antibiotics normalized by $B^{K-}$ with 50 µg/mL Kan) is fit to an exponential decay curve such that $\frac{\%B_0^K}{\%B_{Kan}^K} = x_1 e^{-x_2 t}$, where the calculated plasmid loss rate constant for plasmid K is $\kappa_K = \frac{x_2}{24}$ h$^{-1}$. Similarly, the same procedure is performed to determine the effect of Pheno on $\kappa_K$, where $\%B^{K-}$ treated with Pheno is used instead of $\%B^{K-}$ without any treatment, e.g.: $\frac{\%B_{Pheno}^K}{\%B_{Kan}^K} = x_1 e^{-x_2 t}$.

**Conjugation dynamics for native plasmids.** For all plasmids tested, those previously characterized were measured using known resistance markers from literature (e.g., R388, R6K, RP4, PCU1, K, and C). ESBL pathogen conjugation ability and resistance spectrum was established in a previous publication[34]. In particular, Carb resistance was transferred from conjugation-capable ESBL donors to $R^0$ recipient for further characterization. R carrying one of the nine tested plasmids is denoted $R^P$ for generality in this description (Supplementary Table 1). Incompatibility groups are determined via MLST courtesy of the Anderson group[75]. Some ESBLs indicate multiple MLST identification, suggesting either fused incompatibility groups or multiple plasmids. Indeed, gel electrophoresis indicates multiple plasmids in recipient strains for these cases (not shown).

All additional plasmids were transferred to R for all other measurements. Growth rates of the additional plasmids $R^P$ were obtained using the same methodology for the engineered system, namely, log-transforming growth curves from a plate reader fitting the linear portion to a regression. $\alpha$ was calculated in the same way as described previously, namely, by normalizing the growth rate of $R^0$ by the growth rate of $R^P$ (Supplementary Fig. 3B). Conjugation efficiency estimates for all native plasmids were performed using $R^0$ as the recipient except for the plasmid C, which used $B^0$ as recipient. For all ESBL conjugation efficiency estimates, the native strains were used as donors. Donors with compatible resistance markers with the plasmid and $R^0$ were used for the remaining plasmids (Supplementary Table 1, Supplementary Fig. 3A). For the long-term experiments, equal volumes of $R^0$ and $R^P$ were mixed and diluted 10,000× from the overnight culture into a 96-well plate. Plates were sealed using both sealing membranes and grown for 23 h at 30 °C or 37 °C (see Supplementary Table 1), and shaking at 250 rpm. Every few days, the percentage of $R^P$ remaining in the mixture was quantified using CFU. The percentage was obtained by spreading diluted culture to a range that resulted in 20–150 countable colonies, which was typically a net of $1 \times 10^7$-fold onto Cm and Cm+Carb plates (but changed depending on the growth rate of the strain being tested). For all plasmid information, see Supplementary Table 1. Identical protocols for the engineered system and native plasmids were used for all inhibition experiments, where the media was supplemented with 3.25 mM Lin or 120 µM Pheno, or both fresh daily (Fig. 4e, Supplementary Fig. 6C).

To initiate the three-species (distinct *E. coli* strains) three-plasmid experiment, six populations ($R^0$, $Y^0$, $B^0$, $R^{R6K}$, $Y^{R388}$, and $B^{RP4}$) were mixed together in arbitrary fractions. The mixture was propagated daily every 24 h at a dilution of 10,000×. To quantify population fraction, flow cytometry was used to distinguish red, yellow and blue fluorescence, and measurements were taken daily. To quantify plasmid fraction every 3–4 days, selective plating was performed to determine the fraction of each individual plasmid (CFU from plates containing both Cm and either Strep, Kan, or Tm divided by CFU obtained from Cm plates). To determine competition between the three populations, $R^0$, $B^0$, and $Y^0$ only were mixed and similarly propagated. No CFU measurements were taken as there were no plasmids present. Note that for this set of experiments, $B^0$ is different from $B^0$ used in the previous ones, as it expresses $Cm^R$ chromosomally instead of $Amp^R$.

**Estimating $\eta_{Crit}$.** The cost of each plasmid was determined relative to $R^0$, the plasmid-free counterpart (Supplementary Figs. 3, 6, and Supplementary Table 3). We assume the plasmid loss rate to be small for all plasmids ($0.001\ h^{-1}$, as with K, Supplementary Fig. 1A). Indeed, large plasmids (like most conjugal, >20 kb)[25] are typically low copy number to minimize burden[60,76,77], and therefore often employ one or more active partitioning mechanisms to promote stable inheritance[78–80]. To compare with $\eta_{Crit}$, $\eta_C$ was determined using $R^0$ as the recipient harvested during stationary phase (Supplementary Fig. 3A, Supplementary Table 3). Similar to K, we account for the physiological influence on $\eta_C$ by using an upper estimate $2.5 \times 10^3$ greater than the measured value for comparison with $\eta_{Crit}$.

**Data availability.** The authors declare that all the relevant data supporting the findings of the study are available in this article or its Supplementary Information files, or from the corresponding author upon request.

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

## Acknowledgements

We thank D.I. Andersson for strain constructs, N. Buchler for access to a flow cytometer, S. Tuncay for flow cytometry training and troubleshooting, D. Anderson for access to the ESBL library, M. Redinbo and T. Dimitriu for plasmid constructs, and T. Sysoeva for insightful comments and suggestions. This study was partially supported by the U.S. Army Research Office under grant #W911NF-14-1-0490 (L.Y.), National Institutes of Health (L.Y.: 2R01-GM098642, 1RO1AI25604), a David and Lucile Packard Fellowship (L.Y.), and the National Science Foundation grant under the Division of Mathematical Sciences #1614538 from the math biology program (R.D.).

## Author contributions

A.J.L. conceived the research, designed and performed both modeling and experimental analyses, interpreted the results, and wrote the manuscript. H.R.M. and J.K.S. assisted in experimental setup, data interpretation, and manuscript revisions. C.P. assisted in performing experiments, data analysis, and manuscript revisions. R.D. assisted in modeling analysis and manuscript revisions. L.Y. conceived the research, assisted in research design, data interpretation, and manuscript writing.

## Additional information

**Competing interests:** The authors declare no competing financial interests.

