## [Peer Review File · Nature Communications]

Reviewers' comments:

Reviewer #1 (Remarks to the Author):

This is a really nice paper using models and simple, elegant experiments to beautifully demonstrate the importance of conjugation dynamics in the maintenance of antibiotic resistance plasmids. The paper is very clear and well written, the modelling and experiments are carefully designed and appear sound, the figures are logical and easy to follow. The findings are of general interest and important for our understanding of plasmid dynamics and the spread of traits like antibiotic resistance by HGT. I have only minor comments:

1. I find the new term CASPER unhelpful and unnecessary; this isn't an entirely new idea, and it doesn't need an acronym. Related there are too many repetitive statements about priority in the paper (e.g. see point 3 for recent similar uncited work on the importance of conjugation for plasmid persistence without selection) - the manuscript presents very nice original work whose novelty speaks for itself!
2. I wonder how general is the finding that individual plasmid dynamics are additive in combination? I would assume it is limited to understanding multiple compatible plasmids, but that dynamics would be more complex for incompatible plasmids? If so, I think the authors should add appropriate caveats
3. You should cite and discuss the recent paper by Stevenson et al ISME J 2017 (http://www.nature.com/ismej/journal/vaop/ncurrent/abs/ismej20174_2a.html) which similarly shows persistence of an environmental mercury resistance plasmid by conjugation in the absence of positive selection
4. Your coverage of the literature on compensatory evolution is out of date and you should cite:
San Millan et al 2014 <https://www.nature.com/articles/ncomms6208>
San Millan 2015 <https://www.nature.com/articles/ncomms7845>
Harrison et al 2015 [http://www.cell.com/current-biology/abstract/S0960-9822\(15\)00718-6](http://www.cell.com/current-biology/abstract/S0960-9822(15)00718-6)
Porse et al 2016 <https://www.ncbi.nlm.nih.gov/pmc/articles/PMC5062321/>
Yano et al 2016 <http://onlinelibrary.wiley.com/doi/10.1111/mmi.13407/abstract>
5. You could consider in your discussion recent work on the interplay between conjugation and compensatory evolution with respect to plasmid stability by Hall et al Plasmid 2017 <https://www.ncbi.nlm.nih.gov/pubmed/28461121>

Reviewer #2 (Remarks to the Author):

In this manuscript the authors combine mathematical modelling and in vitro experiments to evaluate the role of horizontal transmission in the stability of plasmids in a population. Understanding the evolutionary forces that maintain costly drug-resistance genes in a population when antibiotics are not used is an important problem that could aid in the design of clinical interventions that reverse drug resistance and, in this context, this study is timely and relevant. I have a few comments that I believe should be addressed prior to publication:

- First, I don't think that introducing an acronym into the literature when referring to a well-known phenomenon is particularly useful. I would recommend the authors avoid using the term CASPER and refer to the phenomenon explicitly as conjugation-assisted persistence.
- The mathematical model presented in this manuscript is a simple compartment model written as a set of ordinary differential equations. In general, I find the notation (based on using $\{0,1\}$ to denote the presence or not of a given plasmid as a superscript) very confusing, particularly as the number of plasmids increases (Equations S16-39 are absolutely impossible to read). I would recommend writing the system of equations in an abstract, general form.
- The numerical simulations presented in this study are performed with varying initial conditions

obtained from a uniform distribution and illustrated in Figures 2 and 3 as multiple lines in the same plot. This is unnecessary, as the steady-state analysis shows that the system presents a single attractor, and thus the fixed point doesn't depend on the initial fractions of each subpopulation. If the authors want to show that their result is robust, then they could have performed simulations of the model with noisy parameters obtained from a distribution with mean the value estimated from the experimental data, or numerically solved a system of stochastic differential equations.

- The model considers a constant dilution rate, $D > 0$, as in a chemostat model. However, their experimental setup considers serial dilutions with daily transfers. Why is this discrepancy? The authors could have modeled this experimental system by considering that the initial condition of each season is a projection of the terminal condition of the previous season. But of course, this would have produced a hybrid dynamical system with growth-dilution cycles that would have complicated considerably the steady-state analysis. Or used a continuous culture device to contrast the theoretical predictions with experimental data.

- The authors argue that fitness costs are a function of the environmental concentration of antibiotic, a valid assumption that is implemented in the model by considering that the parameter α can present different values, depending on the drug concentration. I think it would have been better if the model considered explicitly α as a function of the antibiotic, instead of a fixed parameter with different numerical simulations using different values. In order to calibrate this parameter, the authors normalize experimental growth rates of plasmid-bearing and plasmid-free cells using different antibiotic concentrations (Fig S1C), but I think it would have been better to quantify cost based on competition experiments with equal initial fractions of the population and then estimating the final relative frequency of each strain using selective media or a flow cytometer.

- By assuming that the system reaches steady state, the authors obtain an expression (Eq 1) for a critical conjugation efficiency as a function of the rate of segregational loss and the fitness cost/benefit associated to plasmid bearing, $\eta_{crit} = \alpha(\kappa + D) - D$. This is a nice, simple expression that is validated with experimental data and represents the main contribution of this study. As the authors mention, this is conceptually similar to the Stewart & Levin criteria for plasmid stability. In lines 103-107 the authors argue that their criteria overcomes the challenges associated with "decoupling plasmid loss measurements from fitness costs". However, in the supplementary information the authors describe the experiments used to calibrate their model parameters and as far as I can tell their population-level estimates of conjugation and segregational loss could also present identifiability issues (Figure S1). For instance, the rate of conjugation is estimated using a previously published protocol that estimates the fractions of donors, recipients and transconjugants in a population using selective media. These values are then used to obtain a conjugation efficiency estimate, η , that depends on cell density and thus is normalized with respect to the carrying capacity. Similarly, to estimate the rate of plasmid segregational loss, κ , the authors propagate a plasmid-carrying population in a serial dilution experiment and estimate a plasmid loss rate constant by fitting an exponentially decreasing function to the obtained time-series of the fraction of plasmid-free cells in the population. Notably, these two parameters, η and κ , are obtained after periods of growth and therefore can be confounded with complex population dynamics resulting from differences in fitness between plasmid-free and plasmid-carrying subpopulations.

- In the discussion (line 305), the authors mention that recent studies have shown that compensatory adaptation can ameliorate the fitness cost associated to plasmid-bearing. Do the authors find evidence of compensatory adaptation in their evolved strains? In any case, it would be interesting to evaluate the role of compensatory mutations in their model, in order to evaluate the conditions when conjugation-assisted persistence is preferred over compensation-assisted persistence.

In summary, I believe this is a well written paper presenting a simple dynamical model with clear

predictions validated with a nice data set. I do believe, however, that the experimental and theoretical results presented in this manuscript represent only a marginal contribution to our understanding of the forces that drive plasmid dynamics. Indeed, the interaction between fitness cost, segregational loss and conjugation under different selection regimes is an interesting and unresolved problem that has received considerable attention in the past few years, but the controversy does not arise, as the authors claim, because "some have suggested that conjugation is not a major mechanism responsible for the persistence of plasmids" (line 279). On the contrary, the consensus in the community since the publication of the seminal studies by Levin & Stewart (and a series of subsequent studies, both theoretical and experimental) is that horizontal transmission is sufficient to maintain a costly plasmid in a population, what Bergstrom et al. (2004) referred to as "hitchhiking" and was re-defined in this paper as CASPER.

Reviewer 1:

1. *“This is a really nice paper using models and simple, elegant experiments to beautifully demonstrate the importance of conjugation dynamics in the maintenance of antibiotic resistance plasmids. The paper is very clear and well written, the modelling and experiments are carefully designed and appear sound, the figures are logical and easy to follow. The findings are of general interest and important for our understanding of plasmid dynamics and the spread of traits like antibiotic resistance by HGT. I have only minor comments”*

We appreciate the reviewer’s recognition of the importance of our findings. We have fully addressed the raised issues below and in our revised manuscript (in blue).

2. *“I find the new term CASPER unhelpful and unnecessary; this isn't an entirely new idea, and it doesn't need an acronym. Related there are too many repetitive statements about priority in the paper (e.g. see point 3 for recent similar uncited work on the importance of conjugation for plasmid persistence without selection) - the manuscript presents very nice original work whose novelty speaks for itself!”*

We appreciate the feedback from the reviewer and have removed the term. We also edited the text to clarify the context of our work to better differentiate it from previous studies. As addressed in point 3, we have updated the citations to reflect more relevant studies.

3. *“I wonder how general is the finding that individual plasmid dynamics are additive in combination? I would assume it is limited to understanding multiple compatible plasmids, but that dynamics would be more complex for incompatible plasmids? If so, I think the authors should add appropriate caveats.”*

We thank the reviewer for raising this very interesting point. We agree these findings are entirely contingent on plasmid compatibility, and more generally, independently existing plasmids (e.g. that cost, conjugation rate, and plasmid loss are not drastically altered by the presence of another plasmid). In fact, if the plasmids are incompatible, then persistence would instead depend on whether the individual strains were capable of coexisting, which would depend on the relative cost and conjugation efficiency of each plasmid alone. Similarly, the additive effect would be negated if the plasmids expressed surface exclusion, which prevents conjugation of one or more plasmids from entry. We have updated the text for emphasis (Page 8-9):

“Moreover, modeling predicts conjugation-assisted persistence to occur for a single species carrying multiple conjugation plasmids. This is contingent on the plasmids’ ability to exist independently of each other (e.g. distinct incompatibility groups to ensure compatible replication machinery and the absence of surface exclusion that prevents entry of one of the plasmids), and the fact that other relevant plasmid parameters (i.e., η_C , α , or κ) are not drastically altered by the presence of another plasmid (Supplementary Methods Eq S11-14).”

4. *“You should cite and discuss the recent paper by Stevenson et al ISME J 2017 (<http://www.nature.com/ismej/journal/vaop/ncurrent/abs/ismej201742a.html>) which similarly shows persistence of an environmental mercury resistance plasmid by conjugation in the absence of positive selection.”*

We thank the reviewer for bringing this to our attention. Indeed, this article is a convincing example of similar conclusions, demonstrating the infectious nature of HGT to facilitate the spread of resistance in various selection environments. We have updated the discussion section with this citation (Page 11-12):

“Our results conclusively demonstrate that the transfer of various conjugal plasmids is sufficiently fast to exhibit persistence. This is consistent with recent work demonstrating the persistence of heavy metal resistance in the absence of positive selection when transferrable via conjugation, compared to inheritable solely through clonal expansion (the latter of which required positive selection to maintain)¹.”

5. *"Your coverage of the literature on compensatory evolution is out of date and you should cite: San Millan et al 2014 <https://www.nature.com/articles/ncomms6208>, San Millan 2015 <https://www.nature.com/articles/ncomms7845>, Harrison et al 2015 [http://www.cell.com/current-biology/abstract/S0960-9822\(15\)00718-6](http://www.cell.com/current-biology/abstract/S0960-9822(15)00718-6), Porse et al 2016 <https://www.ncbi.nlm.nih.gov/pmc/articles/PMC5062321/>, Yano et al 2016, <http://onlinelibrary.wiley.com/doi/10.1111/mmi.13407/abstract>."*

We appreciate the reviewer for their suggestions. Indeed, these studies are great examples of potential ways in which bacteria enhance their own plasmid stability by modulating the key parameters we (and others) have identified. In fact, these findings coupled with our own work, represents an exciting future synergistic direction in determining the extent to which such mutations reduce fitness cost alone, or potentially modulate conjugation efficiency and/or plasmid segregation as well. We have updated the discussion section to include a more in-depth discussion of this idea, and have incorporated each of the suggested citations (Page 12).

"Several other factors may contribute to the long-term stability of such plasmids in the environment. Indeed, almost half of all plasmids are unable to transfer via conjugation. Co-evolution between the host and plasmid can compensate for fitness cost²⁻⁶, and recent work has shown that other factors such as positive selection coupled with compensatory adaptation can help explain long-term plasmid persistence^{5,7}. Interestingly, compensatory mutations may also modulate other key processes involved with mobile genetic element (MGE) upkeep by directly or indirectly modulating processes involved in conjugation, such as decreased expression of MGE replication⁶, down regulation of global gene expression⁸, or increased plasmid copy number⁷. Indeed, recent theoretical work emphasized the potentially paradoxical role of these interacting processes, where reducing fitness cost could mitigate the evolution for higher conjugation rates⁹. Future work investigating the extent to which compensatory mutations individually modulate α , κ , η_C , or some combination thereof, would provide critical insight into predicting evolutionary trajectories that enhance plasmid stability."

6. *"You could consider in your discussion recent work on the interplay between conjugation and compensatory evolution with respect to plasmid stability by Hall et al Plasmid 2017."*

We appreciate the reviewer's suggestion, and find that this study fits in nicely with the ones suggested in the previous comment. Indeed, there is a complicated relationship between α , κ , η_C , and the paradoxical findings in this study emphasize this notion. This work has been incorporated into the section highlighted in our response to the previous comment.

Reviewer 2:

1. *"Understanding the evolutionary forces that maintain costly drug-resistance genes in a population when antibiotics are not used is an important problem that could aid in the design of clinical interventions that reverse drug resistance and, in this context, this study is timely and relevant."*

We thank the reviewer for recognizing the importance of our study. The reviewer raised some comments regarding modeling justification and establishing the appropriate context of our study. We found these suggestions highly helpful, and have addressed them fully below and in our revised manuscript.

2. *"First, I don't think that introducing an acronym into the literature when referring to a well-known phenomenon is particularly useful. I would recommend the authors avoid using the term CASPER and refer to the phenomenon explicitly as conjugation-assisted persistence."*

We appreciate the reviewer's suggestion and have removed the term from the current version.

3. *“The mathematical model presented in this manuscript is a simple compartment model written as a set of ordinary differential equations. In general, I find the notation (based on using (0,1) to denote the presence or not of a given plasmid as a superscript) very confusing, particularly as the number of plasmids increases (Equations S16-39 are absolutely impossible to read). I would recommend writing the system of equations in an abstract, general form.”*

We thank the reviewer for pointing this out. To derive a general structure, we let N be the number of species, p be the number of plasmids, such that there are $N \times 2^p$ total populations (all species/plasmid combinations). Note that conjugation can be algorithmically implemented by matrix operations between vectors of unique sets of plasmids. To facilitate writing the equation, we define the following set of definitions and well-defined linear algebra operations:

Let $z = 2^p - 1$ be the number of unique plasmid combinations indexed by $0 \leq j \leq z = 2^p - 1$. Then we define $\gamma_j = [b_1 \ b_2 \ \dots \ b_p]$, $\forall j$ be the row vector consisting of binary elements such that $b_i = \begin{cases} 1 & \text{plasmid} \\ 0 & \text{no plasmid} \end{cases}$. Therefore, the set of all γ_j ($\{\gamma_j\}$) consists of the ordered set of all possible unique plasmid combinations represented as binary vectors. For example, γ_0 is the first vector in $\{\gamma_j\}$ that consists of all zeros (no plasmids), and γ_z is the last vector in $\{\gamma_j\}$ and represents a vector of all ones. Thus, $S_n^{\gamma_0}$ represents the n -th species that carries no plasmids, and $S_n^{\gamma_z}$ represents the n -th species that carries all plasmids at once.

Then we can define the following basic linear algebra operators:

1. Let $\bar{\gamma}_j = 1 - \gamma_j$ be the complement of γ_j .
2. Let $H(\gamma_i, \gamma_k) = \gamma_j \cdot \gamma_k = \sum_{\phi=1}^p \gamma_i[\phi] \gamma_k[\phi]$ be the dot product.

Several useful operations follow:

1. $H(\gamma_j, \gamma_j)$ is the identity operation, and will give the sum of all 1's in γ_j . Intuitively, this calculates the number of unique plasmids in a single population. For example, let $\gamma_j = [0 \ 1 \ 1]$:

$$H(\gamma_j, \gamma_j) = [0 \ 1 \ 1] \cdot [0 \ 1 \ 1] = (0 \cdot 0) + (1 \cdot 1) + (1 \cdot 1) = 2.$$

2. $H(\gamma_j, \gamma_k)$ is the number of elements shared between two vectors. Intuitively, this represents the number of unique plasmids carried between two populations. For example, let $\gamma_j = [0 \ 1 \ 0]$ and $\gamma_k = [0 \ 1 \ 1]$:

$$H(\gamma_j, \gamma_k) = [0 \ 1 \ 0] \cdot [0 \ 1 \ 1] = 1.$$

3. $H(\bar{\gamma}_j, \gamma_k)$ is the number of elements in γ_j that are not in γ_k . Intuitively, this represents the number of plasmids that are in one population, but not the other. Thus, this operation calculates the number of potential conjugation interactions between two sets of plasmids γ_j and γ_k . For example:

$H(\bar{\gamma}_j, \gamma_k) = [1 \ 0 \ 1] \cdot [0 \ 1 \ 1] = 1$. γ_k can therefore donate a single plasmid to γ_j . Say instead, $\gamma_j = [1 \ 0 \ 0]$. Then $H(\bar{\gamma}_j, \gamma_k) = [0 \ 1 \ 1] \cdot [0 \ 1 \ 1] = 2$. This means a population carrying γ_k can donate one of two plasmids to the recipient carrying γ_j .

Using these rules, the differential equation for a unique population $\frac{dS_n^{\gamma}}{dt}$ can be generally represented in the following equation, and is broken down into five individual terms in the table below:

$$\frac{dS_n^{\gamma}}{dt} = \mu_n^{\gamma} S_n^{\gamma} \left(1 - \frac{(\sum_{i=1}^N \sum_{j=0}^z S_i^j)}{N_m} \right) + \eta_c \sum_{\forall k} S_n^{\gamma_k} \sum_{i=1}^N \sum_{\forall \rho} S_i^{\gamma_{\rho}} - S_n^{\gamma} \eta_c \sum_{i=1}^N \sum_{\forall q} S_i^{\gamma_q} H(\bar{\gamma}, \gamma_q) + \kappa \sum_{\forall \lambda} S_n^{\gamma_{\lambda}} - \kappa S_n^{\gamma} H(\gamma, \gamma) - D S_n^{\gamma}$$

Term	Representation	Additional definitions and description
Logistic growth	$\mu_n^Y S_n^Y \left(1 - \frac{(\sum_{i=1}^N \sum_{j=0}^z S_i^j)}{N_m} \right)$	There are a total of $N \times 2^p$ populations
Contribution from conjugation	$\eta_c \sum_{\forall k} S_n^{Y_k} \sum_{i=1}^N \sum_{\forall \rho} S_i^{Y_\rho}$	Recipients: Define k to be the indices corresponding to a subset of vectors in $\{\gamma_j\}$ ($\{\gamma_k\} \subseteq \{\gamma_j\}$) such that each vector has one unique plasmid difference to γ , and has less total plasmids (number of plasmids carried by $S_n^{Y_k}$ is less than the number of plasmids carried by S_n^Y for all k). Donor: Define ρ to be the indices corresponding to a subset of vectors in $\{\gamma_j\}$ consisting of every vector combination that carries the missing plasmid as defined by γ_k .
Loss from conjugation	$-S_n^Y \eta_c \sum_{i=1}^N \sum_{\forall q} S_i^{Y_q} H(\bar{\gamma}, \gamma_q)$	Recipients: The population S_n^Y will gain a plasmid through conjugation Donor: Define q to be the indices corresponding to a subset of vectors in $\{\gamma_j\}$ that has at least one unique plasmid not contained by S_n^Y .
Contribution from plasmid segregation	$\kappa \sum_{\forall \lambda} S_n^{Y_\lambda}$	Define λ to be the indices corresponding to a subset of vectors in $\{\gamma_j\}$ consisting of every vector combination that has every single plasmid in γ plus exactly one more plasmid ($\{\gamma + 1\}$).
Loss from plasmid segregation	$-\kappa S_n^Y H(\gamma, \gamma)$	S_n^Y loses each of the plasmids in γ with the rate constant κ
Dilution	$-D S_n^Y$	First order kinetics for dilution

Due to the complexity, we prefer not to include this in the main text (unless the reviewer feels strongly otherwise), and have left the notation 0 and 1 for the all cases involving $n=1,2$ and $p=1,2$. As the reviewer noted, $n=3, p=3$ already becomes fairly complex.

In light of the reviewer's comment, we recognize the need to improve our presentation. To this end, we have included a diagram to facilitate the interpretation of our model equations (Supplementary Fig. 5).

4. The following two points are addressed simultaneously:

- *“The numerical simulations presented in this study are performed with varying initial conditions obtained from a uniform distribution and illustrated in Figures 2 and 3 as multiple lines in the same plot. This is unnecessary, as the steady-state analysis shows that the system presents a single attractor, and thus the fixed point doesn't depend on the initial fractions of each subpopulation. If the authors want to show that their result is robust, then they could have performed simulations of the model with noisy parameters obtained from a distribution with mean the value estimated from the experimental data, or numerically solved a system of stochastic differential equations.”*
- *“The model considers a constant dilution rate, $D > 0$, as in a chemostat model. However, their experimental setup considers serial dilutions with daily transfers. Why is this discrepancy? The authors could have modeled this experimental system by considering that the initial condition of each season is a projection of the terminal condition of the previous season. But of course, this would have produced a hybrid dynamical system with growth-dilution cycles that would have complicated considerably the steady-state analysis. Or used a continuous culture device to contrast the theoretical predictions with experimental data.*

We appreciate the reviewer's suggestion and agree that the initial conditions are not critical for the overall conclusion in the simplest model. Our reasoning for this was based on our observations that the total population size decreased daily slightly until around the 3rd day, indicating that our experimental parameters

were operating in such a way that dilution was faster than growth to the carrying capacity, which simulates quasi chemostat-like environment. We acknowledge that this was an assumption, and our presented modeling is not entirely consistent with the precise experimental setup.

To address the reviewer's comments, we implemented the model to be entirely consistent with experimental setup to investigate whether changing D influenced the overall dynamics. Here, generating randomly chosen initial conditions is consistent with the variability introduced experimentally by a strong initial dilution (~80 cells/well) across many well replicates.

Modeling demonstrates that the particular implementation of D does not significantly influence the qualitative results and overall conclusions of our study (e.g. modeling with periodic dilution, Fig. S1D). Both continuous and discrete models are consistent with our experimental results. We have included the comparison between two versions of the model in the Supplemental Information. Correspondingly, we have updated the main text to clarify this point (Page 6-7), and described in the supplement (Page 5):

“We note that our model assumes a constant dilution rate constant (D), which represents an approximation of the discrete, periodic dilutions in our experiments. Simulations using a model implementing discrete dilutions generated qualitatively the same results (Supplementary Fig. 1D)”

5. *“The authors argue that fitness costs are a function of the environmental concentration of antibiotic, a valid assumption that is implemented in the model by considering that the parameter α can present different values, depending on the drug concentration. I think it would have been better if the model considered explicitly α as a function of the antibiotic, instead of a fixed parameter with different numerical simulations using different values. In order to calibrate this parameter, the authors normalize experimental growth rates of plasmid-bearing and plasmid-free cells using different antibiotic concentrations (Fig S1C), but I think it would have been better to quantify cost based on competition experiments with equal initial fractions of the population and then estimating the final relative frequency of each strain using selective media or a flow cytometer.”*

We appreciate this interpretation of our modeling parameter. We agree that competition experiments are the typical standard for measuring fitness cost over time between two competing populations in many cases. We would like to clarify the rationale of our approach and note that our long-term experiments can indeed be considered competition experiments:

- The growth rate of a population (alone or in a mixture) serves as a reliable proxy for overall fitness¹⁰. It also represents a fundamental parameter that dictates the dynamics of microbial communities. Our definition of α as the ratio of growth rates is consistent with previous seminal work in the field (as the reviewer mentioned) by Levin and Stewart, who also use the monoculture growth rates to estimate fitness burden of the plasmid itself, and others investigating burden associated with resistance mechanisms^{11,12}. This is particularly suited for estimating fitness using our synthetic system, since the strains are otherwise identical, and the fully characterized plasmids do not encode any elements imparting an active mechanism for competition. In our system, we do not expect that the plasmid-carrying and plasmid-free populations to have such higher-order interactions.
- Estimating α using growth rate estimates are critical to maintain consistency between different native plasmids, because unlike the synthetic system, we do not have non-conjugating plasmid controls. Therefore, estimating α via competition would be confounded by conjugation, and undermine the purpose of our study.
- Fitness estimates based on long-term competition are a lumped trait that incorporates contribution of many factors, including growth rates of different populations, additional interactions between different populations, and initial population structure. Our flow-cytometer data using the synthetic system (Fig. 2B i) in the absence of conjugation and antibiotic selection is indeed the type of competition experiment the reviewer suggested. Our results are fully consistent with estimating α based on the more fundamental growth rate parameters.

6. *“By assuming that the system reaches steady state, the authors obtain an expression (Eq 1) for a critical conjugation efficiency as*

a function of the rate of segregational loss and the fitness cost/benefit associated to plasmid bearing, $\eta_{crit} = \alpha(\kappa + D) - D$. This is a nice, simple expression that is validated with experimental data and represents the main contribution of this study. As the authors mention, this is conceptually similar to the Stewart & Levin criteria for plasmid stability. In lines 103-107 the authors argue that their criteria overcomes the challenges associated with "decoupling plasmid loss measurements from fitness costs". However, in the supplementary information the authors describe the experiments used to calibrate their model parameters and as far as I can tell their population-level estimates of conjugation and segregational loss could also present identifiability issues (Figure S1). For instance, the rate of conjugation is estimated using a previously published protocol that estimates the fractions of donors, recipients and transconjugants in a population using selective media. These values are then used to obtain a conjugation efficiency estimate, η , that depends on cell density and thus is normalized with respect to the carrying capacity. Similarly, to estimate the rate of plasmid segregational loss, κ , the authors propagate a plasmid-carrying population in a serial dilution experiment and estimate a plasmid loss rate constant by fitting an exponentially decreasing function to the obtained time-series of the fraction of plasmid-free cells in the population. Notably, these two parameters, η and κ , are obtained after periods of growth and therefore can be confounded with complex population dynamics resulting from differences in fitness between plasmid-free and plasmid-carrying subpopulations."

We thank the reviewer for raising this very important point. Indeed, the parameter estimation is a critical component of this work. We address these points individually:

Conjugation efficiency:

The conjugation efficiency is approximated using a bimolecular representation of the conjugation process ($D+R \rightarrow T$), where D, R and T respectively represent donor, recipient, and transconjugant. Assuming that neither D nor R changes over Δt , we can derive the explicit equation for η_c , or the conjugation efficiency, where $\eta_c = T / (DR\Delta t)^{13}$. Previously, we demonstrated that the negligible growth assumption is valid under conjugation conditions of one-hour incubation, minimal media (M9), and room temperature ($^{\circ}25C$)¹⁴. In our current study, we used the identical conditions to eliminate the potential inference of selection dynamics on the estimates of η_c .

Segregation loss:

To estimate segregation loss, the reviewer is correct in that our measurement is done as it is traditionally implemented, by quantifying the observed decline in plasmid-carrying cells over time in the absence of selection. As the reviewer noted, this measurement is subjected to bias due to cost. However, this bias can be reduced by using a plasmid with a minimal effect on the growth rate. Indeed, our mathematical analysis has shown that the observed plasmid loss rate deviates from the true loss rate by $\Delta_p = \mu_1(\alpha - 1) + \frac{\ln(x_1)}{t}$ ($\kappa_{obs} = \kappa + \Delta_p$, calculations below). According to our estimates, for $x_1 = 0.96$ (flow cytometer detection limit, see Supplementary Fig. 2A) and small time $t = 10 \text{ hr}^{-1}$ (consistent with estimates of α), then $\Delta_p \approx 0.0019$, which is less than the standard error associated with fitting the plasmid loss rate (≈ 0.0022) (see Supplementary Fig. 1A). Therefore, the difference between κ_{obs} (e.g. cost confounded) and true κ under these conditions is less than the error associated with the experimental technique. Indeed, changing the modeling estimate of κ_{obs} (an upper estimate) by a factor of 5, which greatly over-estimates the potential variability in the parameter, insignificantly changes our overall conclusion.

We have updated the text to overall increase clarity as far as the precise ways in which our quantification reduces the influence of confounding factors. We have also updated the main text (Page 4) and supplement (Page 2-4) with this specific mathematical analysis to emphasize this notion:

“According to Eq 1, a plasmid will be maintained as long as the conjugation efficiency is sufficiently fast compared with the rate of plasmid loss and fitness burden (Fig. 1C-D). Beyond maintenance, sufficiently fast conjugation efficiency is necessary for the plasmid-carrying population to be dominant ($S' > S^0$, Fig. 1C), even when a plasmid is slightly beneficial (α is slightly < 1). Our criterion is similar to that derived by Stewart and Levin¹², but avoids experimental challenges associated with decoupling plasmid loss measurements from

fitness cost¹⁵. Experimentally, observed plasmid loss (κ_{Obs}) can be determined by measuring the time constant of decay for a non-transferrable plasmid, which represents a combined effect of true κ and α (Supplementary Fig. 1A)¹⁶. Indeed, analysis shows that $\kappa_{obs} \approx \kappa$ for plasmids with minimal fitness effects ($\alpha \approx 1$). Since these two parameters are challenging to decouple, our criterion lumps the effects of α and κ together. To determine κ_{obs} , we chose to use a low-cost plasmid ($\alpha = 1.02$) to minimize the confounding effects of cost. Based on our experimentally determined parameters, analysis shows the standard error associated with fitting the plasmid loss rate (≈ 0.0022) is greater than the difference between κ_{obs} and κ (see Supplementary Methods, Plasmid Loss Calculations, for complete derivation).”

“Plasmid Loss Calculations (Pages 2-3, Supplementary Methods)

Experimentally, we measure plasmid loss by propagating a population initiated with 100% plasmid-carrying cells that are non-transferrable, and quantifying the percentage of plasmid-carrying cells over time. The decay of the fraction of plasmid carrying cells (B^1) is measured using the following equation:

$$\frac{B^1}{B^1+B^0} = x_1 e^{-x_2 t}$$

where the rate constant x_2 represents the *observed* plasmid loss rate (κ_{obs}). Since the plasmid is non-transferrable, conjugation does not confound these measurements. However, since the plasmid is costly, κ_{obs} is a function of the cost as well, since competition cannot be excluded.

To determine the extent to which cost confounds the plasmid loss rate constant, consistent with our experimentally setup we assume:

1. Cells grow approximately exponentially
2. $\eta_C = 0$

The differential equations can be written as follows:

$$\begin{aligned} \frac{dB^1}{dt} &= \mu_1 B^1 - B^1 \kappa - D B^1 \\ \frac{dB^0}{dt} &= \alpha \mu_1 B^0 + B^1 \kappa - D B^0 . \end{aligned}$$

This system of differential equations can be solved analytically, such that:

$$\begin{aligned} B^1 &= e^{\lambda t}, \text{ where } \lambda = \mu_1 - \kappa - D, \text{ and} \\ B^0 e^{t(D-\alpha\mu_1)} &= \int \kappa e^{t(\lambda+D-\alpha\mu_1)} dt \\ B^0 &= \frac{\kappa}{D-\alpha\mu_1+\lambda} e^{t(\lambda+D-\alpha\mu_1)} + C_1 e^{t(\alpha\mu_1-D)} \\ C_1 &= -\frac{\kappa}{D-\alpha\mu_1+\lambda} \\ B^0 &= \frac{\kappa}{D-\alpha\mu_1+\lambda} (e^{t\lambda} - e^{t(\alpha\mu_1-D)}) \end{aligned}$$

From the solution of B^0 , we can see that if the plasmid has a low cost (e.g. $\alpha \approx 1$, which is true for plasmid K ($\alpha = 1.02$)) then

$$B^0 \approx (e^{t(\alpha\mu_1-D)} - e^{t\lambda})$$

Therefore, substituting B^1 and B^0 with the explicit form, we have:

$$\begin{aligned} \ln(B^1) - \ln(B^1 + B^0) &= \ln(x_1) - x_2 t \\ t(\lambda + x_2) &= \ln(x_1) + \ln(e^{t\lambda} - e^{t\lambda} + e^{t(\alpha\mu_1-D)}) \end{aligned}$$

This gives us the final equation, where $x_2 = \kappa_{obs}$:

$$\kappa_{obs} = \kappa + \mu_1(\alpha - 1) + \frac{\ln(x_1)}{t}$$

This shows that κ_{obs} is most accurate as $\alpha \rightarrow 1$ (low cost), e.g. $\kappa_{obs} \approx \kappa$. As the cost increases, κ_{obs} is confounded by the growth rate. In that case, $\kappa = \kappa_{obs} + \Delta_p$, where $\Delta_p = \mu_1(\alpha - 1) + \frac{\ln(x_1)}{t}$. According to our estimates, for $x_1=0.96$ (flow detection limit) and small time (e.g. $t=10$ for relevant range of α) then $\Delta_p \approx 0.0019$, which is less than the error associated with estimating κ_{obs} (≈ 0.0022).”

7. *“In the discussion (line 305), the authors mention that recent studies have shown that compensatory adaptation can ameliorate the fitness cost associated to plasmid-bearing. Do the authors find evidence of compensatory adaptation in their evolved strains? In any case, it would be interesting to evaluate the role of compensatory mutations in their model, in order to evaluate the conditions when conjugation-assisted persistence is preferred over compensation-assisted persistence.”*

We agree that compensatory mutations raise interesting questions that warrant follow-up work. Indeed, several studies have demonstrated the contribution of compensatory adaptation to plasmid stability. The extent to which such mutations contribute to plasmid stability by modulating one or a combination of fitness cost, conjugation efficiency, or plasmid loss is an intriguing question that currently remains unresolved.

In our study, we do not find evidence of compensatory evolution in the data, since individual colonies were chosen and re-propagated and the dynamics appeared similar. Though we cannot rule out the possibility entirely, its contribution in the time scale of our experiments is limited. For instance, the contribution from potential compensatory evolution, if it had occurred, was insufficient to maintain the model plasmid in the absence of conjugation (Fig. 2B). Yet, conjugation was able to maintain the plasmid without any lag.

In light of reviewer’s comment, we used modeling to investigate the extent to which compensatory evolution could influence the overall results. To do this, we introduced a mutant population (M) that is generated from the plasmid-carrying population at a constant rate β . Spontaneous mutations arise at a frequency of around 10^{-9} per base pair per generation; of these, only a fraction will confer increased resistance. We assume an unusually high mutation rate ($\beta = 10^{-5} \text{ hr}^{-1}$)^{17,18}, and examine the scenarios where the mutation either neutralizes the plasmid cost, or confers a fitness benefit (e.g. $\alpha = 1$, or 0.9 , respectively, Supplementary Fig. 3F-G). Regardless of the α , the plasmid was eliminated in the absence of conjugation, and it did not significantly alter the temporal dynamics in the presence of conjugation.

In all, our experimental data and further modeling analysis suggest that compensatory mutations of this low-cost plasmid likely did not contribute significantly to the overall dynamics. This analysis has been included in the supplementary methods under the section titled “One-species one-plasmid model” (Page 6).

“To examine the potential influence of compensatory adaptation, we modified this model to include a third population of mutants, M:

$$\begin{aligned} \frac{dS^0}{dt} &= \alpha\mu S^0(1 - S^0 - S^1 - M) - \eta_c S^1 S^0 + \kappa(M + S^1) - DS^0, \\ \frac{dS^1}{dt} &= \mu S^1(1 - S^0 - S^1 - M) + \eta_c S^1 S^0 - \kappa S^1 - DS^1 - \beta S^1, \\ \frac{dM}{dt} &= \alpha_M \mu S^1(1 - S^0 - S^1 - M) + \eta_c M S^0 - \kappa M - DM + \beta S^1. \end{aligned} \quad S5$$

S^1 transitions to M at a rate β , such that M has reduced fitness burden and grows at a rate equal to S^0 . Analysis demonstrates that regardless of whether the plasmid can conjugate, a rapid transition rate ($\beta \approx 0.001$ and above) will result in a significant portion of the final plasmid population

consisting of M compared to S¹ after 14 days (Supplementary Fig. 2B). This value is large compared to typical estimates of mutation rates. Particularly, it is estimated that spontaneous mutations arise at a frequency of around 10⁻⁹ per base pair per generation¹⁷, and only a fraction of these will confer increased resistance. In our study, the plasmid was readily eliminated in the absence of conjugation (Fig. 2B). Thus, this analysis suggests that compensatory mutations of this low-cost plasmid likely did not contribute significantly to the overall dynamics.”

8. *“...but the controversy does not arise, as the authors claim, because “some have suggested that conjugation is not a major mechanism responsible for the persistence of plasmids” (line 279). On the contrary, the consensus in the community since the publication of the seminal studies by Levin & Stewart (and a series of subsequent studies, both theoretical and experimental) is that horizontal transmission is sufficient to maintain a costly plasmid in a population, what Bergstrom et al. (2004) referred to as “hitchhiking” and was re-defined in this paper as CASPER.”*

We agree that studies from Levin & Stewart, along with those that closely followed, were foundational literature in establishing the role of HGT, and some (but not all) follow-up studies have suggested that plasmids can be maintained via conjugation. However, we would like to respectfully point out several references since the seminal work by Stewart and Levin that illustrate points of contention or confusion, underscoring the scope of our work in the overall context of plasmid persistence. This confusion is part due to the lack of quantitative measurements of various parameters, including the conjugation efficiency. Taken together, the majority of past studies in fact tend to question the feasibility of plasmid maintenance by conjugation alone.

Below is a comprehensive list to illustrate the wide breadth of conclusions and debate over the role of conjugation in propagating resistance in the absence of selection since the Levin & Stewart publication (emphasis added by us):

- Simonsen et al (1991)¹⁹: “These results suggest that the rate of plasmid transfer in natural populations of *E. coli* is far too low for plasmids to be maintained as parasitic DNA.”
- Bergstrom, et al (2000)²⁰: The model assumes that “plasmids cannot persist simply by bearing genes that are beneficial to their hosts” and instead invoked the ability of plasmids to “hitchhike” on the adaptive bacterial mutants responsible for selective sweeps or to transfer selective genes between populations adapted to different niches.
- Lili et al (2007)²¹: “Second, *in vitro* estimates of transfer rates, population density, segregation rates, and fitness costs, have suggested that infectious transfer *per se* is insufficient for plasmid maintenance), who’s work expanded on the Levin criteria by demonstrating a wider basis for plasmid persistence.”
- Smillie, et al (2010)²²: “Since half of the plasmids are nonmobilizable, including the largest ones, these results seriously question the purely parasitic nature of all plasmids, which require high rates of transfer (13, 91),” where references 13 and 91 refer to Bergstrom et al and Lili et al, respectively.
- San millan, et al (2014)⁵: “Nonetheless, the transfer conditions required for plasmids to act as pure genetic parasites are very restrictive and most of the previous studies (but not all) reject this hypothesis based on theoretical considerations.”
- Normann et al (2009)²³: “Conjugation therefore also contributes to plasmid stability since, theoretically, it rids the population of intercellular competition as long as transfer rates are higher than cell division rates.”
- Rankin, et al (2011)²⁴: “Stewart and Levin (1977) built one of the first models to analyse plasmid persistence, and argued that plasmids could not persist under low rates of HGT. This has provoked a debate as to whether horizontal transfer is sufficient for mobile elements to persist in populations because of their costs on host fitness (see, for example, Bergstrom et al., 2000; Lili et al., 2007; Slater et al., 2008)...yet the question of **how parasitic mobile elements can persist remains an open**

question...”

- Schluter, et al (2007)²⁵: “The question as to whether or not this horizontal spread can allow plasmids to persist as parasitic genetic elements without ever benefiting their host, is currently unanswered. Stewart and Levin (Stewart & Levin, 1977) and later studies (Bergstrom et al., 2000) reported that the transfer rates required for plasmid R1, measured in chemostats, were not high enough for the plasmid to be parasitic. Very recent results with IncP-1 plasmids on solid surface however suggest the opposite.”
- MacLean, et al (2015)²⁶: “Although plasmids carry genes that can potentially benefit their bacterial hosts, it remains challenging to understand how plasmids can persist in bacterial populations over the long term — an evolutionary dilemma that has been called the ‘**plasmid paradox**’ [7]... **Recurrent horizontal gene transfer** of plasmids between bacteria **could theoretically resolve the plasmid paradox** and allow plasmids to stably persist in bacterial populations [10]. However, **the general consensus is that rates of plasmid transfer are simply too low for horizontal transfer to maintain plasmids in the absence of selection** for plasmid-carried genes [11]. For example, whole-genome sequencing has shown that only about 50% of plasmids are capable of transmitting themselves horizontally via conjugation [12]; the remaining 50% must rely on alternative mechanisms”

More so, several studies have demonstrated that conjugal plasmids are either maintained, or eliminated, in the absence of selection. Plasmid persistence was attributed to fast conjugation, low/no cost, or both, whereas plasmid elimination was attributed to the opposing processes (e.g. slow conjugation, high burden, or both). Examples of these are found below:

Persistence

- Bahl, et al (2007)²⁷: “Results show that the plasmid’s ability to conjugate counteracts plasmid loss and is thus an important mechanism for the stable maintenance of IncP-1 plasmids within the gastrointestinal environment.”
- Cottell, et al (2012)²⁸: “In conclusion, plasmids such as pCT have evolved to impose little impact on host strains. Therefore, the persistence of antibiotic resistance genes and their vectors is to be expected in the absence of antibiotic selective pressure regardless of antibiotic stewardship.”

Elimination

- Fox, et al (2008)²⁹: “The **plasmid was unable to invade in liquid**. When carbon source levels were lower or not replenished, plasmid invasion was hampered. Simulations of the mathematical model closely matched the experimental results and produced estimates of the effects of alternative experimental parameters. This allowed us to isolate the likely mechanisms most responsible for the observations. In conclusion, **spatial structure and nutrient availability can be key determinants in the invasiveness of plasmids**.” To summarize, this study invoked spatial distribution as a key contributor to persistence since conjugation alone could not maintain the plasmid in liquid culture.
- Subbiah, et al (2011)³⁰: “Competition studies showed that carriage of *bla*_{CMY-2} **plasmids imposed a measurable fitness cost** on the host bacteria both *in vitro* (0.095 to 0.25) and *in vivo* (dairy calf model). Long-term passage experiments in the absence of antibiotics demonstrated that plasmids with limited antibiotic resistance phenotypes arose, but eventually drug-sensitive, plasmid-free clones dominated the populations. Given that plasmid decay or loss is inevitable, we infer that **some level of selection is required for the long-term persistence** of *bla*_{CMY-2} plasmids in bacterial populations.”

Different outcomes likely depend on inter-experimental differences including the plasmid type and

mating procedure. Presumably, these outcomes can all be reconciled by accurately quantifying the underlying parameters driving plasmid persistence. However, lack of quantitative experiments has prevented general conclusions.

In light of reviewers' comments, we have expanded our introduction and discussion to clarify our findings in the current literature landscape.

References

- 1 Stevenson, C., Hall, J. P. J., Harrison, E., Wood, A. J. & Brockhurst, M. A. Gene mobility promotes the spread of resistance in bacterial populations. *ISME J*, doi:10.1038/ismej.2017.42 (2017).
- 2 Händel, N., Otte, S., Jonker, M., Brul, S. & ter Kuile, B. H. Factors That Affect Transfer of the IncI1 β -Lactam Resistance Plasmid pESBL-283 between *E. coli* Strains. *PLoS ONE* **10**, doi:10.1371/journal.pone.0123039 (2015).
- 3 Harrison, E. *et al.* Rapid compensatory evolution promotes the survival of conjugative plasmids. *Mobile Genetic Elements* **6**, e1179074, doi:10.1080/2159256X.2016.1179074 (2016).
- 4 Dionisio, F., Conceição, I. C., Marques, A. C. R., Fernandes, L. & Gordo, I. The evolution of a conjugative plasmid and its ability to increase bacterial fitness. *Biology Letters* **1**, 250-252, doi:10.1098/rsbl.2004.0275 (2005).
- 5 San Millan, A. *et al.* Positive selection and compensatory adaptation interact to stabilize non-transmissible plasmids. *Nat Commun* **5**, 5208, doi:10.1038/ncomms6208 (2014).
- 6 San Millan, A., Toll-Riera, M., Qi, Q. & MacLean, R. C. Interactions between horizontally acquired genes create a fitness cost in *Pseudomonas aeruginosa*. *Nat Commun* **6**, 6845, doi:10.1038/ncomms7845 (2015).
- 7 Yano, H. *et al.* Evolved plasmid-host interactions reduce plasmid interference cost. *Mol Microbiol* **101**, 743-756, doi:10.1111/mmi.13407 (2016).
- 8 Harrison, E., Guymer, D., Spiers, A. J., Paterson, S. & Brockhurst, M. A. Parallel compensatory evolution stabilizes plasmids across the parasitism-mutualism continuum. *Current biology : CB* **25**, 2034-2039, doi:10.1016/j.cub.2015.06.024 (2015).
- 9 Hall, J. P. J., Brockhurst, M. A., Dytham, C. & Harrison, E. The evolution of plasmid stability: Are infectious transmission and compensatory evolution competing evolutionary trajectories? *Plasmid* **91**, 90-95, doi:10.1016/j.plasmid.2017.04.003 (2017).
- 10 Wisner, M. J. & Lenski, R. E. A Comparison of Methods to Measure Fitness in *Escherichia coli*. *PLoS ONE* **10**, e0126210, doi:10.1371/journal.pone.0126210 (2015).
- 11 Mariam, D. H., Mengistu, Y., Hoffner, S. E. & Andersson, D. I. Effect of *rpoB* Mutations Conferring Rifampin Resistance on Fitness of *Mycobacterium tuberculosis*. *Antimicrobial Agents and Chemotherapy* **48**, 1289-1294, doi:10.1128/aac.48.4.1289-1294.2004 (2004).
- 12 Stewart, F. M. & Levin, B. R. The population biology of bacterial plasmids: a priori conditions for the existence of conjugationally transmitted factors. *Genetics* **87**, 209-228 (1977).
- 13 Lopatkin, A. J., Sysoeva, T. A. & You, L. Dissecting the effects of antibiotics on horizontal gene transfer: Analysis suggests a critical role of selection dynamics. *Bioessays* **38**, 1283-1292, doi:10.1002/bies.201600133 (2016).

- 14 Lopatkin, A. J. *et al.* Antibiotics as a selective driver for conjugation dynamics. *Nature Microbiology*, 16044, doi:10.1038/nmicrobiol.2016.44 (2016).
- 15 Lau, B. T. C., Malkus, P. & Paulsson, J. New quantitative methods for measuring plasmid loss rates reveal unexpected stability. *Plasmid* **70**, 353-361, doi:10.1016/j.plasmid.2013.07.007 (2013).
- 16 Bernard, A. & Payton, M. Selection of Escherichia coli expression systems. *Current protocols in protein science* **Chapter 5**, Unit5.2, doi:10.1002/0471140864.ps0502s00 (2001).
- 17 De Gelder, L. *et al.* Combining mathematical models and statistical methods to understand and predict the dynamics of antibiotic-sensitive mutants in a population of resistant bacteria during experimental evolution. *Genetics* **168**, 1131-1144, doi:10.1534/genetics.104.033431 (2004).
- 18 Schulz zur Wiesch, P., Engelstädter, J. & Bonhoeffer, S. Compensation of Fitness Costs and Reversibility of Antibiotic Resistance Mutations. *Antimicrobial Agents and Chemotherapy* **54**, 2085-2095, doi:10.1128/aac.01460-09 (2010).
- 19 Simonsen, L. The existence conditions for bacterial plasmids: Theory and reality. *Microb Ecol* **22**, 187-205, doi:10.1007/BF02540223 (1991).
- 20 Bergstrom, C. T., Lipsitch, M. & Levin, B. R. Natural Selection, Infectious Transfer and the Existence Conditions for Bacterial Plasmids. *Genetics* **155**, 1505-1519 (2000).
- 21 Lili, L. N., Britton, N. F. & Feil, E. J. The persistence of parasitic plasmids. *Genetics* **177**, 399-405, doi:10.1534/genetics.107.077420 (2007).
- 22 Smillie, C., Garcillán-Barcia, M. P., Francia, M. V., Rocha, E. P. C. & Cruz, F. d. I. Mobility of Plasmids. *Microbiol Mol Biol Rev* **74**, 434-452, doi:10.1128/MMBR.00020-10 (2010).
- 23 Norman, A., Hansen, L. H. & Sørensen, S. J. Conjugative plasmids: vessels of the communal gene pool. *Philos Trans R Soc Lond B Biol Sci* **364**, 2275-2289, doi:10.1098/rstb.2009.0037 (2009).
- 24 Rankin, D. J., Rocha, E. P. C. & Brown, S. P. What traits are carried on mobile genetic elements, and why? *Heredity* **106**, 1-10, doi:10.1038/hdy.2010.24 (2011).
- 25 Schlüter, A., Szczepanowski, R., Pühler, A. & Top, E. M. Genomics of IncP-1 antibiotic resistance plasmids isolated from wastewater treatment plants provides evidence for a widely accessible drug resistance gene pool. *FEMS Microbiology Reviews* **31**, 449-477, doi:10.1111/j.1574-6976.2007.00074.x (2007).
- 26 MacLean, R. C. & San Millan, A. Microbial Evolution: Towards Resolving the Plasmid Paradox. *Current biology : CB* **25**, R764-767, doi:10.1016/j.cub.2015.07.006 (2015).
- 27 Bahl, M. I., Hansen, L. H., Licht, T. R. & Sorensen, S. J. Conjugative transfer facilitates stable maintenance of IncP-1 plasmid pKJK5 in Escherichia coli cells colonizing the gastrointestinal tract of the germfree rat. *Appl Environ Microbiol* **73**, 341-343, doi:10.1128/aem.01971-06 (2007).
- 28 Cottell, J. L., Webber, M. A. & Piddock, L. J. V. Persistence of Transferable Extended-Spectrum- β -Lactamase Resistance in the Absence of Antibiotic Pressure. *Antimicrobial Agents and Chemotherapy* **56**, 4703-4706, doi:10.1128/AAC.00848-12 (2012).
- 29 Fox, R. E., Zhong, X., Krone, S. M. & Top, E. M. Spatial structure and nutrients promote invasion of IncP-1 plasmids in bacterial populations. *ISME J* **2**, 1024-1039, doi:10.1038/ismej.2008.53 (2008).
- 30 Subbiah, M., Top, E. M., Shah, D. H. & Call, D. R. Selection Pressure Required for Long-Term Persistence of blaCMY-2-Positive IncA/C Plasmids \square . *Applied and Environmental Microbiology* **77**, 4486-4493, doi:10.1128/AEM.02788-10 (2011).

REVIEWERS' COMMENTS:

Reviewer #2 (Remarks to the Author):

I've read the revised version of the manuscript "Persistence and reversal of plasmid-mediated antibiotic resistance" and believe the manuscript has substantially increased in clarity. In particular, I found the table presented in the rebuttal describing the terms of the model very useful and, although I agree with the authors that including this abstract description of the model in the main text would hinder its readability, I think it should be included in the methods or supplementary information. In summary, the authors have addressed many of my concerns and therefore I recommend this paper for publication.

Reviewer 2:

“I’ve read the revised version of the manuscript "Persistence and reversal of plasmid-mediated antibiotic resistance" and believe the manuscript has substantially increased in clarity. In particular, I found the table presented in the rebuttal describing the terms of the model very useful and, although I agree with the authors that including this abstract description of the model in the main text would hinder its readability, I think it should be included in the methods or supplementary information. In summary, the authors have addressed many of my concerns and therefore I recommend this paper for publication.”

We are extremely grateful that the reviewer believes our updated manuscript has addressed all previous concerns raised. Based on the reviewer’s suggestion, we have included the general modeling form and table for each contributing term in the supplement (Section titled ‘Generalized conjugation model’ and Supplementary Table 4).